# ALIGNDIFF: EXPLOITING MODEL-INTRINSIC SIGNALS FOR BETTER PREFERENCE DATA SELECTION

## ABSTRACT

Aligning large language models with human preferences remains challenging, and the quality of preference data is critical for effective alignment. Existing large-scale datasets often introduce noise and distribution shifts, limiting model performance. To address this, we propose AlignDiff, a preference data filtering framework driven by intrinsic model signals. AlignDiff first identifies samples with clear preferences using both positive and inverse signals, then prioritizes the more challenging samples based on the average negative log-likelihood gap, encouraging the model to learn richer information from them. Across multiple models and benchmarks, AlignDiff consistently outperforms the other seven baselines. On AlpacaEval 2.0, training on only 50% of the data selected by AlignDiff nearly doubles the performance of LLaMA-3-8B-SFT compared to training on the full dataset. The data filtered by AlignDiff preserves the length gap distribution while achieving a more favorable external reward margins distribution, and difficulty-based curriculum learning further enhances model performance.

## 1 INTRODUCTION

As large language models (LLMs) advance, aligning outputs with human expectations and mitigating risks remains a major challenge (Zhao et al., 2025). Early post-training efforts relied on preference alignment algorithms such as RLHF (Ouyang et al., 2022) and DPO (Rafailov et al., 2023), trained on preference data, to guide model behavior toward human values (Wang et al., 2023a). As research progresses, reinforcement fine-tuning (e.g., using the GRPO algorithm (Shao et al., 2024)) enhances model reasoning and alignment, potentially reducing reliance on traditional preference alignment. However, Lanchantin et al. (2025) recently found that semi-online DPO can match the performance of fully online GRPO while substantially decreasing training costs, indicating that further research on DPO is valuable. As the core of DPO, preference data provides models with more reliable preference signals and lays a solid foundation for subsequent method improvements (Meng et al., 2024; Ethayarajh et al., 2024). Early studies typically trained DPO directly on large-scale preference datasets such as UltraFeedback (Cui et al., 2023), aiming to cover a broad human-preference space in one shot. However, follow-up work (Xiao et al., 2025; Shen et al., 2024) has shown that simply "piling on data" introduces noise and distribution shifts that degrade alignment performance, highlighting the need to filter the data (Wang et al., 2024) carefully.

To address this issue, Previous work (Morimura et al., 2024; Hu et al., 2024b) has proposed various preference data filtering methods that are grounded in external signals (e.g., scores produced by LLM-as-a-judge), leveraging external reward scores (Yasunaga et al., 2024; Pattnaik et al., 2024) or data attributes (Yu et al., 2025). However, those methods may introduce preference data that does not align with the model's intrinsic learning preferences (e.g., data with very low implicit reward margins under DPO), which can negatively affect the stability of alignment training (Xu et al., 2024; Wang et al., 2025). To mitigate these issues, recent studies have increasingly leveraged the implicit rewards of DPO for preference data filtering (Hu et al., 2024b; Kim et al., 2025; Chen et al., 2025). Nevertheless, such methods still exploit only a small portion of the model's internal information, leaving many latent signals unexplored. This limitation not only constrains their effectiveness but also exacerbates the "squeezing effect" in DPO optimization: training on samples with the highest implicit reward margin tends to further shrink the high-likelihood regions of both chosen and rejected responses, thereby biasing the model toward generating outputs that are neither clearly chosen nor clearly rejected (Ren & Sutherland, 2025).

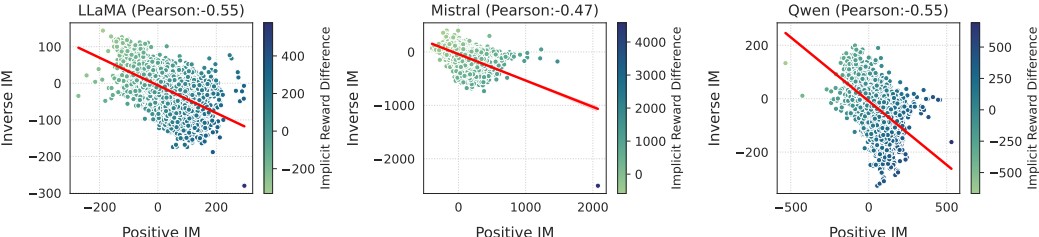

Figure 1: *Complementarity between inverse and positive preference signals.* We present the scatter plot of the joint distribution of positive and inverse implicit reward margins (IM) computed by SFT models on UltraFeedback samples. The results show that inverse IM and positive IM do not exhibit a clear inverse correlation, indicating that *inverse signals can provide additional information not captured by positive signals, helping to understand the model's preference distribution better*.

In this work, we propose **AlignDiff**, a preference data filtering framework driven by intrinsic model signals. It integrates Mining Clearer Preference Samples via **Align**ment Discrepancy and Sample **Diff**iculty-aware Calibration to select a high-quality preference dataset. Our key observation is that standard DPO can exploit positive and inverse preference signals by swapping chosen and rejected responses, thus providing complementary information. Motivated by this, we first filter raw preference pairs using alignment discrepancy computed based on positive and negative implicit reward margins to identify samples with clear preferences. However, some samples in this dataset are simplistic for the model, yielding negligible training benefit, and may even induce a "squeezing effect" (Ren & Sutherland, 2025): it further shrinks the high-likelihood regions of both chosen and rejected responses. To address these issues, we calibrate the dataset through the lens of sample difficulty: estimating difficulty via the average negative log-likelihood gap and prioritizing more challenging samples. Extensive experiments demonstrate the effectiveness of AlignDiff. On AlpacaEval 2.0 (Dubois et al., 2024), it consistently outperforms existing preference data selection baselines, achieving a 3.9% improvement in length-controlled performance while maintaining comparable output length to that of the best baseline SDPO (Hu et al., 2024b). Training on only the top 50% of data selected by our method nearly doubles the performance of LLaMA-3-8B-SFT compared to training on the full dataset. The advantages of AlignDiff are likewise observed on MT-Bench (Zheng et al., 2023). Moreover, the dataset obtained by AlignDiff preserves the length gap distribution (ensuring that observed improvements stem from enhanced preference quality rather than length-related bias) while achieving a more favorable external reward margin distribution. A difficulty-based curriculum learning can further boost model performance by building on the high-quality dataset.

## 2 BACKGROUND

To better understand our method, we first review the core ideas of Reinforcement Learning from Human Feedback (Ouyang et al., 2022), which naturally leads to the implicit reward mechanism employed in Direct Preference Optimization (Rafailov et al., 2023), the foundation of our method.

RLHF is a widely used framework for aligning LLMs with human preferences via preference learning. Let $\mathcal{D}$ be a dataset consisting of pairwise comparisons $(\mathbf{x}, \mathbf{y}_w, \mathbf{y}_l)$, where $\mathbf{y}_w$ is the preferred response over $\mathbf{y}_l$ for a given prompt $\mathbf{x}$, RLHF trains a reward model $r_\phi(\mathbf{y} \mid \mathbf{x})$ using the Bradley–Terry model (Bradley & Terry, 1952) and optimizes it by minimizing the cross-entropy loss. The policy $\pi_\theta$ is refined via RL algorithms such as PPO (Schulman et al., 2017) by maximizing the expected reward, with a KL constraint $\lambda$ enforcing the learned policy to stay close to the reference policy $\pi_{\text{ref}}$:

$$\mathcal{J}_{\text{RLHF}}(\boldsymbol{\theta}) = \mathbb{E}_{\mathbf{x}\sim\mathcal{D}, \mathbf{y}\sim\pi_\theta(\cdot|\mathbf{x})} \left[ r_\phi(\mathbf{y}|\mathbf{x}) \right] - \lambda \cdot \mathbb{D}_{\text{KL}} \left[ (\pi_\theta(\cdot|\mathbf{x})\|\pi_{\text{ref}}(\cdot|\mathbf{x})) \right]. \quad (1)$$

DPO simplifies this process by eliminating the reward model. It directly leverages preference data by optimizing a contrastive loss over the preference dataset $\mathcal{D}$. Let $\pi_\theta$ be the policy model and $\pi_{\text{ref}}$ the reference model (e.g., a supervised fine-tuned model), the DPO loss is:

$$\mathcal{L}_{\text{DPO}}(\pi_\theta, \mathcal{D}) = -\mathbb{E}_{(\mathbf{x},\mathbf{y}_w,\mathbf{y}_l)\sim\mathcal{D}} \left[ \log \sigma \left( \beta \log \frac{\pi_\theta(\mathbf{y}_w|\mathbf{x})}{\pi_{\text{ref}}(\mathbf{y}_w|\mathbf{x})} - \beta \log \frac{\pi_\theta(\mathbf{y}_l|\mathbf{x})}{\pi_{\text{ref}}(\mathbf{y}_l|\mathbf{x})} \right) \right], \quad (2)$$

where $\sigma$ is the sigmoid function, and $\beta$ is a hyperparameter controlling logit-scale sensitivity. This loss implicitly defines a reward function, commonly called the "implicit reward":

$$r_{\text{implicit}}(\mathbf{y}|\mathbf{x}) = \log \frac{\pi_\theta(\mathbf{y}|\mathbf{x})}{\pi_{\text{ref}}(\mathbf{y}|\mathbf{x})} = \log \pi_\theta(\mathbf{y}|\mathbf{x}) - \log \pi_{\text{ref}}(\mathbf{y}|\mathbf{x}). \quad (3)$$

Figure 2: ***Overview of the AlignDiff Framework***. We leverage Alignment Discrepancy ($R_{AD}$), which utilizes both positive and inverse preference signals to identify samples with clear preferences, and then select high-difficulty data based on the Average Negative Log-Likelihood Gap (ANG) to improve data quality and mitigate the squeezing effect. AlignDiff relies solely on the model's internal information, ensuring that the selected data better aligns with the model's inherent preferences.

By leveraging log probabilities as implicit preference signals, DPO avoids explicit reward modeling, simplifying the optimization process and improving its stability. Specifically, it models preference between a preferred response $\mathbf{y}_w$ and a less preferred response $\mathbf{y}_l$ given input $\mathbf{x}$ as:

$$p(\mathbf{y}_w \succ \mathbf{y}_l | \mathbf{x}) = \sigma\left(\beta \log \frac{\pi_{\boldsymbol{\theta}}(\mathbf{y}_w|\mathbf{x})}{\pi_{\text{ref}}(\mathbf{y}_w|\mathbf{x})} - \beta \log \frac{\pi_{\boldsymbol{\theta}}(\mathbf{y}_l|\mathbf{x})}{\pi_{\text{ref}}(\mathbf{y}_l|\mathbf{x})}\right). \tag{4}$$

In essence, the optimization objective defined by this preference modeling approach is equivalent to maximizing the **implicit reward margin (IM)**:

$$\text{M}_{\text{im}}(\mathbf{y}_w, \mathbf{y}_l | \mathbf{x}) = r_{\text{implicit}}(\mathbf{y}_w|\mathbf{x}) - r_{\text{implicit}}(\mathbf{y}_l|\mathbf{x}). \tag{5}$$

A larger margin indicates clearer preferences and correlates with faster convergence during training (Rafailov et al., 2023). Consequently, several works (Chen et al., 2024; Deng et al., 2025) have proposed using IM as a criterion to identify high-quality preference data.

## 3 ALIGNDIFF

We aim to extract a high-quality subset from a preference dataset that has already been filtered. In this work, we propose the AlignDiff framework (Figure 2), leveraging the model's internal information to perform high-quality preference data selection. Inspired by the symmetry of the DPO loss, we propose Alignment Discrepancy (§3.2), which utilizes positive and inverse preference signals to identify samples with clear preferences further. Furthermore, to encourage the model to learn higher-value information during training, we propose calibrating the dataset from the perspective of sample difficulty (§3.3), prioritizing the selection of more challenging samples. The details of the AlignDiff are shown in the algorithm 1.

### 3.1 MOTIVATION

We find that the IM is inherently tied to modeling positive preference signals in standard DPO. Interestingly, due to its symmetry, the DPO objective can also be used to model negative preference signals when the chosen and rejected responses are swapped in Eq. 4. This insight leads to a natural question: *Can we leverage both positive and inverse preference signals to facilitate the selection of higher-quality preference data?* Formally, we define two policy models: the positive preference policy $\pi_{\boldsymbol{\theta}}^{\text{positive}}$, trained to align with preferred responses via the standard DPO loss, and the inverse preference policy $\pi_{\boldsymbol{\theta}}^{\text{inverse}}$, trained on inverse preference data to prefer typically dispreferred responses, with a DPO loss that swaps the roles of the preferred and dispreferred responses:

$$\mathcal{L}_{\text{DPO}}^{+}\left(\pi_{\boldsymbol{\theta}}^{\text{positive}}, \mathcal{D}\right) = -\mathbb{E}_{(\mathbf{x},\mathbf{y}_w,\mathbf{y}_l)\sim\mathcal{D}}\left[\log \sigma\left(\beta \log \frac{\pi_{\boldsymbol{\theta}}^{\text{positive}}(\mathbf{y}_w|\mathbf{x})}{\pi_{\text{ref}}(\mathbf{y}_w|\mathbf{x})} - \beta \log \frac{\pi_{\boldsymbol{\theta}}^{\text{positive}}(\mathbf{y}_l|\mathbf{x})}{\pi_{\text{ref}}(\mathbf{y}_l|\mathbf{x})}\right)\right], \tag{6}$$

$$\mathcal{L}_{\text{DPO}}^{-}\left(\pi_{\boldsymbol{\theta}}^{\text{inverse}}, \mathcal{D}\right) = -\mathbb{E}_{(\mathbf{x},\mathbf{y}_w,\mathbf{y}_l)\sim\mathcal{D}}\left[\log \sigma\left(\beta \log \frac{\pi_{\boldsymbol{\theta}}^{\text{inverse}}(\mathbf{y}_l|\mathbf{x})}{\pi_{\text{ref}}(\mathbf{y}_l|\mathbf{x})} - \beta \log \frac{\pi_{\boldsymbol{\theta}}^{\text{inverse}}(\mathbf{y}_w|\mathbf{x})}{\pi_{\text{ref}}(\mathbf{y}_w|\mathbf{x})}\right)\right]. \tag{7}$$

These two policy models induce fundamentally different reward behaviors: while $\pi_{\boldsymbol{\theta}}^{\text{positive}}$ encourages alignment with human preferences, $\pi_{\boldsymbol{\theta}}^{\text{inverse}}$ exhibits systematically inverted tendencies. The differences between the implicit reward margins induced by the two models provide valuable signals for analyzing preference polarity and understanding model behavior. It can be observed from Figure 1 that the implicit reward margin $\text{M}_{\text{im}}^{\text{inverse}}(\mathbf{y}_w, \mathbf{y}_l|\mathbf{x})$ computed by $\pi_{\boldsymbol{\theta}}^{\text{inverse}}$ does not exhibit a strong negative correlation with $\text{M}_{\text{im}}^{\text{positive}}(\mathbf{y}_w, \mathbf{y}_l|\mathbf{x})$ computed by $\pi_{\boldsymbol{\theta}}^{\text{positive}}$. **This suggests that inverse preference signals are not merely the opposite of positive ones but imply that they capture complementary aspects of preference information.** As empirically validated in §4.3, integrating both signals yields superior performance compared to relying on either signal in isolation.

## 3.2 Mining Clearer Preference Samples via Alignment Discrepancy

Motivated by the observation that the implicit reward margins from $\pi_{\boldsymbol{\theta}}^{\text{inverse}}$ and $\pi_{\boldsymbol{\theta}}^{\text{positive}}$ are not strongly negatively correlated, as well as the underlying discrepancy between these two policy models, we define the Alignment Discrepancy $R_{\textbf{AD}}$ as the difference between the margins $\text{M}_{\text{im}}^{\text{positive}}(\mathbf{y}_w, \mathbf{y}_l|\mathbf{x})$ and $\text{M}_{\text{im}}^{\text{inverse}}(\mathbf{y}_w, \mathbf{y}_l|\mathbf{x})$, computed respectively from $\pi_{\boldsymbol{\theta}}^{\text{positive}}$ and $\pi_{\boldsymbol{\theta}}^{\text{inverse}}$:

$$R_{\text{AD}}(\mathbf{y}_w, \mathbf{y}_l|\mathbf{x}) = \log \frac{\pi_{\boldsymbol{\theta}}^{\text{positive}}(\mathbf{y}_w \mid \mathbf{x})}{\pi_{\boldsymbol{\theta}}^{\text{positive}}(\mathbf{y}_l \mid \mathbf{x})} - \log \frac{\pi_{\boldsymbol{\theta}}^{\text{inverse}}(\mathbf{y}_w \mid \mathbf{x})}{\pi_{\boldsymbol{\theta}}^{\text{inverse}}(\mathbf{y}_l \mid \mathbf{x})}. \tag{8}$$

The detailed derivation can be found in the Appendix I.1. Compared to traditional methods relying only on positive signals, $R_{\text{AD}}$ leverages implicit reward margins from both $\pi_{\boldsymbol{\theta}}^{\text{positive}}$ and $\pi_{\boldsymbol{\theta}}^{\text{inverse}}$ to better capture preference consistency and detect anomalies. Therefore, based on the properties of $R_{\text{AD}}$, we can perform a more refined filtering of the original data and fully leverage it. Specifically, we first define a labeling function $\phi : \mathbb{R} \to \{-1, 0, 1\}$ that categorizes each sample into one of three preference types. Given a threshold $\tau > 0$, $\phi(R_{\text{AD}}(\mathbf{y}_w, \mathbf{y}_l|\mathbf{x}); \tau)$ is defined as:

$$\phi(R_{\text{AD}}(\mathbf{y}_w, \mathbf{y}_l|\mathbf{x}); \tau) = \begin{cases} 1, & \text{if } R_{\text{AD}}(\mathbf{y}_w, \mathbf{y}_l|\mathbf{x}) > \tau \\ 0, & \text{if } -\tau \leq R_{\text{AD}}(\mathbf{y}_w, \mathbf{y}_l|\mathbf{x}) \leq \tau \ , \\ -1, & \text{if } R_{\text{AD}}(\mathbf{y}_w, \mathbf{y}_l|\mathbf{x}) < -\tau \end{cases} \tag{9}$$

where the label $\phi$ encodes model preference polarity based on $R_{\text{AD}}(\mathbf{y}_w, \mathbf{y}_l|\mathbf{x})$: $\phi = -1$ indicates a clear inverse preference, where the model favors the rejected response $\mathbf{y}_l$, potentially due to annotation noise or inconsistencies; $\phi = 0$ denotes an ambiguous case with weak or uncertain model preference; and $\phi = 1$ represents a clear positive preference, where the model's choice aligns with the original human annotation. Then we can get new sample $(\mathbf{x}^{(i)}, \tilde{\mathbf{y}}_w^{(i)}, \tilde{\mathbf{y}}_l^{(i)})$ by processing each old preference pair $(\mathbf{x}^{(i)}, \mathbf{y}_w^{(i)}, \mathbf{y}_l^{(i)}) \in \mathcal{D}$ using the function $\phi$ and the transformation function $\mathcal{F}$:

$$(\mathbf{x}^{(i)}, \tilde{\mathbf{y}}_w^{(i)}, \tilde{\mathbf{y}}_l^{(i)}) = \mathcal{F}(\mathbf{x}^{(i)}, \mathbf{y}_w^{(i)}, \mathbf{y}_l^{(i)}; \phi^{(i)}) = \begin{cases} (\mathbf{x}^{(i)}, \mathbf{y}_l^{(i)}, \mathbf{y}_w^{(i)}), & \text{if } \phi^{(i)} = -1 \\ (\mathbf{x}^{(i)}, \mathbf{y}_w^{(i)}, \mathbf{y}_l^{(i)}), & \text{if } \phi^{(i)} = 1 \\ \varnothing, & \text{if } \phi^{(i)} = 0 \end{cases} . \tag{10}$$

We fully leverage both clear-preference and annotation-affected data to construct a higher-quality dataset $\tilde{\mathcal{D}}$ enriched with explicit preference information:

$$\tilde{\mathcal{D}} = \{(\mathbf{x}^{(i)}, \tilde{\mathbf{y}}_w^{(i)}, \tilde{\mathbf{y}}_l^{(i)}) : \mathcal{F}(\mathbf{x}^{(i)}, \mathbf{y}_w^{(i)}, \mathbf{y}_l^{(i)}; \phi^{(i)}) \neq \varnothing, \forall (\mathbf{x}^{(i)}, \mathbf{y}_w^{(i)}, \mathbf{y}_l^{(i)}) \in \mathcal{D}\}. \tag{11}$$

## 3.3 Sample Difficulty-Aware Dataset Calibration

We have constructed a new preference dataset $\tilde{\mathcal{D}}$ with clear and explicit preferences based on Alignment Discrepancy, which helps the model quickly capture the fundamental alignment direction (Rafailov et al., 2023). However, some samples may be readily distinguishable by the model, offering limited learning value and thus marginal training benefits. The underlying reason is that, in these samples, the chosen response often corresponds to a high-probability output that the model is naturally inclined to generate, while the rejected response corresponds to a very low-probability output disfavored by the model, making further reinforcement of the preference relation less significant. Moreover, as noted by Ren & Sutherland (2025), optimizing these samples with low-probability rejected responses may further trigger the "squeezing effect": it further shrinks the high-likelihood

regions of both chosen and rejected responses (the rapid decrease in chosen and rejected rewards, as shown in Figure 6b and 6c, illustrates this effect), leading the model to generate outputs that are neither clearly chosen nor clearly rejected (Pal et al., 2024; Lan et al., 2025). Therefore, to address these issues, we consider calibrating the preference dataset by selectively filtering from the perspective of sample difficulty: *an ideal preference dataset should keep the chosen responses sufficiently challenging to give the model room for improvement during training, while keeping rejected responses relatively easy to prevent them from being overly suppressed in the DPO phase, thereby avoiding the exacerbation of the "squeezing effect".*

To measure sample difficulty more robustly, we propose the Average Negative Log-Likelihood Gap (ANG), which is less sensitive to response length than Perplexity (PPL), whose exponential formulation amplifies the effect of response length. We first define the average negative log-likelihood (AvgNLL) of a response $\mathbf{y}$ of length $T$ given the input $\mathbf{x}$ as $\overline{\mathrm{NLL}}(\mathbf{y}) = -\frac{1}{T}\sum_{t=1}^{T}\log P(y_t|y_{<t}, \mathbf{x})$. Then, ANG is formulated as the gap between the AvgNLL of the $\mathbf{y}_l$ and that of the $\mathbf{y}_w$:

$$\mathrm{ANG}(\mathbf{y}_w, \mathbf{y}_l) = \overline{\mathrm{NLL}}(\mathbf{y}_w) - \overline{\mathrm{NLL}}(\mathbf{y}_l). \tag{12}$$

A larger (or positive) ANG indicates greater difficulty, as the chosen response $\mathbf{y}_w$ is less likely than the rejected response $\mathbf{y}_l$. Conversely, a smaller (or negative) ANG reflects better alignment and higher fluency of $\mathbf{y}_w$, but provides limited learning signals and may contribute to the "squeezing effect" in DPO. Since the amount of data after AD-filtering varies across models, we select the Top-K preference pairs with the largest ANG to construct the final preference dataset $\mathcal{D}_{\mathrm{final}}$.

## 4 EXPERIMENTS

### 4.1 EXPERIMENTAL SETTINGS

**Base Models and Datasets.** We employ three SFT models as reference backbones for subsequent data filtering and DPO training: LLaMA-3-8B-SFT[1] (Meng et al., 2024), Mistral-7B-SFT[2] (Tunstall et al., 2023), and Qwen-2.5-7B-SFT[3] (Hu et al., 2024b). We adopt the preference dataset UltraFeedback_Binarized (Cui et al., 2023), which has been extensively utilized in alignment research (Pattnaik et al., 2024; Tunstall et al., 2023).

**Evaluation.** To comprehensively evaluate the performance of the model fine-tuned with DPO, we first adopt the AlpacaEval 2.0 (Dubois et al., 2024), which effectively measures the response quality and the alignment capability of the LLMs. We maintain the same annotation methodology (weighted win rate) and report both **WR** (win rate) and the **LC** (length-controlled win rate) to provide a comprehensive assessment of the quality of model generation. We further evaluate the model on MT-Bench (Zheng et al., 2023) to assess its multi-turn dialogue capabilities and overall performance. For more detailed evaluation settings, please refer to the Appendix F.

**Baselines.** To comprehensively evaluate the effectiveness of our method, we conduct an extensive comparison against multiple strong baselines (details of all baselines are provided in Appendix H):

▷ **External Reward Margin (EM)** (Yasunaga et al., 2024; He et al., 2025): As a common setup, we use Qwen2.5-72B-Instruct (Qwen-Team, 2024) to score chosen/rejected samples on four dimensions. We first take the average score across all dimensions as the final score, then calculate the reward margin, and select the samples with the largest margin as high-quality data. See Appendix L for the evaluation template.

▷ **Implicit Reward Margin (IM)** (Chen et al., 2024; Kim et al., 2024): We use the SFT model as the reference model and train on UltraFeedback_Binarized (Cui et al., 2023). Then, we compute the IM for all data in this dataset using the SFT model, and select the data with the largest IM.

▷ **PPLGap**: Based on the characteristics of PPL in reflecting difficulty, we select the preference pair with the largest gap between the chosen PPL and rejected PPL as a high-quality pair.

▷ **Longest-Chosen Preference Pair (LCPP)**: Based on observations in instruction data selection (Shen, 2024), responses with the longest chosen length tend to yield better performance. Therefore, we adopt the preference pairs with the longest chosen responses as a baseline.

---

[1] https://huggingface.co/princeton-nlp/Llama-3-Base-8B-SFT

[2] https://huggingface.co/HuggingFaceH4/mistral-7b-sft-beta

[3] https://huggingface.co/glorgao/Qwen2.5-7B-SFT

▷ **External & Implicit Reward Margin (IM&EM)**: We aggregate the external reward margin and the implicit reward margin following the method described in Deng et al. (2025), and select the data with the largest combined reward margin as high-quality samples.

▷ **R.I.P** (Yu et al., 2025): This method requires selecting preference pairs with the longest rejected responses, while ensuring a sufficient external reward margin.

▷ **SelectiveDPO (SDPO)** (Hu et al., 2024b): SDPO identifies sample difficulty via six reference models, removes hard samples, and applies preference alignment training only to easy ones.

**Implement details.** Following previous works (Lee et al., 2024; Hu et al., 2024a), we adopt the *OpenRLHF* (Hu et al., 2024a) framework to perform DPO training, strictly following the standard procedure. The detailed training procedure is provided in the Appendix G. We use the SFT models as reference models and further train $\pi_\theta^{\text{inverse}}$ and $\pi_\theta^{\text{positive}}$ to conduct more refined data filtering. To ensure a fair comparison across all baseline methods, we standardize the data size to 30k for each.

## 4.2 MAIN RESULTS

Table 1: ***Performance comparison on AlpacaEval 2.0 and MT-Bench using DPO-trained models with various data subsets***. The base models are LLaMA-3-8B-SFT, Mistral-7B-SFT, and Qwen2.5-7B-SFT. The best-performing subsets are highlighted in **bold**.

| Method | LLaMA-3-8B-SFT | | | | Mistral-7B-SFT | | | | Qwen2.5-7B-SFT | | | |
| | Alpaca Eval 2.0 | | | MT-Bench | Alpaca Eval 2.0 | | | MT-Bench | Alpaca Eval 2.0 | | | MT-Bench |
| | LC (%) | WR (%) | Len | Score | LC (%) | WR (%) | Len | Score | LC (%) | WR (%) | Len | Score |
|---|---|---|---|---|---|---|---|---|---|---|---|---|
| Init | 1.2 | 2.2 | 3,841 | 5.2* | 7.6 | 4.4 | 976 | 4.8* | 5.3 | 4.7 | 1,230 | 5.7 |
| Full | 13.7 | 16.8 | 3,431 | 6.5* | 22.3 | 19.4 | 1,713 | 5.9* | 21.3 | 18.9 | 1,715 | 6.8 |
| PPLGAP | 3.7 | 6.3 | 4,002 | 6.5 | 14.4 | 14.4 | 1,931 | 5.6 | 21.2 | 16.9 | 1,631 | 6.7 |
| LCPP | 14.6 | 20.1 | 5,138 | 7.1 | 14.2 | 15.8 | 2,011 | 6.5 | 20.0 | 22.6 | 2,050 | 7.2 |
| EM | 12.6 | 13.5 | 2,076 | 7.0 | 20.8 | 18.8 | 1,801 | **6.8** | 21.7 | 19.4 | 1,740 | 7.2 |
| IM | 19.1 | 19.8 | 2,428 | 7.1 | 27.9 | 26.9 | 1,902 | **6.8** | 27.8 | 27.0 | 1,928 | **7.3** |
| IM&EM | 16.2 | 17.6 | 2,592 | 7.1 | 26.1 | 24.3 | 1,863 | 6.7 | 25.5 | 25.3 | 1,951 | **7.3** |
| R.I.P | 15.9 | 20.0 | 3,461 | 7.1 | 26.4 | 24.5 | 1,862 | **6.8** | 25.1 | 24.3 | 1,902 | **7.3** |
| SDPO | 20.1 | 21.3 | 2,501 | 6.9 | 28.4 | 27.1 | 1,979 | **6.8** | 30.2 | 28.8 | 1,902 | 7.2 |
| **AlignDiff** | **26.4** | **29.3** | 2,680 | **7.2** | **30.6** | **27.6** | 1,852 | **6.8** | **33.4** | **33.8** | 2,035 | **7.3** |

Results marked with * in the table are from SDPO.

**Implicit Rewards Consistently Outperform External Rewards.** Our main results are documented in Table 1. Although EM and IM exhibit low linear correlation, the preference data subset selected by the IM&EM method underperforms compared to that selected using IM alone. Models trained on IM&EM-filtered data achieve lower LC on AlpacaEval 2.0, averaging 1.96% less than those trained on IM-selected data. On MT-Bench, the average score of IM&EM is also slightly lower than that of IM. This demonstrates the challenge of effectively combining IM and EM. Therefore, we recommend placing greater emphasis on IM.

**AlignDiff Outperforms Other Baselines.** AlignDiff selects the highest quality data compared to other baselines. When the average length is comparable to the SDPO method, LC outperforms SDPO by 3.9% on average. It can be observed that by using our method to filter out only 50% of the high-quality data from the original dataset to train LLaMA-3-8B-SFT, the model's performance on Alpaca Eval 2.0 improved nearly twofold compared to the model trained on the full original dataset. Moreover, the average scores on MT-Bench are also superior to those of other baselines. This further demonstrates that the filtered data improves the model's alignment capability.

## 4.3 HYPERPARAMETER AND ABLATION STUDIES

**Hyperparameters.** AlignDiff introduces two key hyperparameters: **(1) Alignment discrepancy threshold** $\tau > 0$ (**Figure 4a**): For Qwen and LLaMA, setting $\tau = 20$ achieves an optimal trade-off between data quality and quantity, leading to the best overall model performance. Since the range of $\tau$ for Mistral differs from that of Qwen and LLaMA due to data quantity constraints, the detailed analysis is provided in the Appendix E. Since the AD range may differ for future models, a similar approach can be adopted: selecting $\tau$ at approximately 5k data points intervals and evaluating their

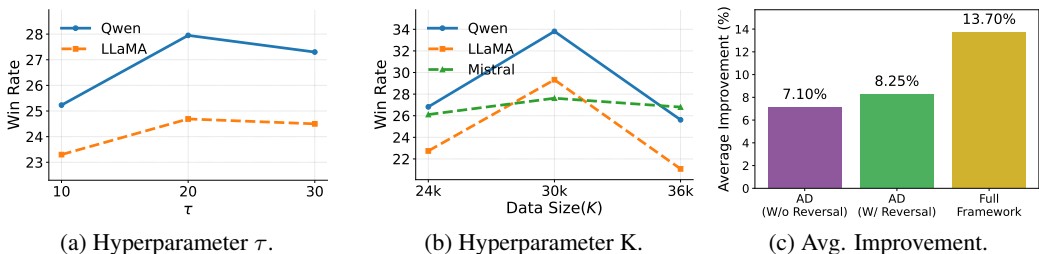

(a) Hyperparameter $\tau$.     (b) Hyperparameter K.     (c) Avg. Improvement.

Figure 4: *Results of Hyperparameter & Ablation Studies*. Panels (a) and (b) show the results of our hyperparameter studies; Panel (c) presents the ablation results for the role of each filtering step.

effects. (2) **Top-K data selection** (**Figure 4b**): The results show that choosing the top 30k samples yields the best performance.

**Comparative Ablation of First-Stage Filtering Methods (Figure 3a)**. We employ AD, Positive IM (PIM), inverse IM (IIM), and EM for first-stage data filtering, conduct DPO training on the Qwen2.5-7B-SFT model using the resulting datasets, and compare the performance of the trained model. The results show that AD outperforms the other methods, demonstrating that combining positive and inverse IM provides richer signals for identifying samples with clearer preferences.

**Validity of Calibration Using Difficult Samples (Figure 3b).** We compare calibration strategies based on easy and random samples, and find that calibration based on difficult samples yields greater performance improvement than both. Among all settings, selecting easy samples for training results in the poorest performance, lagging behind the difficult-sample variant by 9.7%. This gap reinforces the notion that challenging samples contain richer signals that better facilitate model improvement.

**Role of Each Filtering Step (Figure 4c).** To assess the contribution of each component in AlignDiff, we conduct a step-by-step ablation study on the LLaMA and Qwen SFT models, incrementally adding key steps. We evaluate the following settings and compare the results with the unprocessed original dataset: **(1) AD (W/o Reversal)**: Applying alignment discrepancy filtering to $\mathcal{D}$, Only retaining samples with $\phi = 1$; **(2) AD (W/ Reversal)**: Applying both alignment discrepancy filtering and the reversal operation, resulting in $\tilde{\mathcal{D}}$; and **(3) Full Frame-**

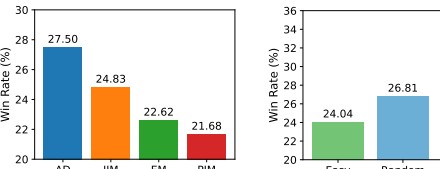

(a) AD vs. other methods    (b) Calib. Strategy Comp.

Figure 3: (a): AD-based data filtering consistently outperforms PIM, IIM, and EM; (b): Calibrating on hard samples boosts performance vs. easy/random baselines.

**work**: Apply the full pipeline, including all components. The results show that model performance consistently improves as we progressively incorporate components of AlignDiff. Introducing alignment discrepancy filtering without reversal (AD (W/o Reversal)) yields a substantial improvement over the unfiltered baseline, and further adding the reversal operation (AD (W/ Reversal)) brings additional gains. AlignDiff combining clear preference and difficulty-aware selection, achieves the best overall performance, highlighting the complementary effects of these components.

## 5 ANALYSIS

### 5.1 A CLOSE LOOK AT THE FILTERED HIGH-QUALITY PREFERENCE DATA

**Optimizing external reward margins while preserving length gap distribution.** In the filtered high-quality preference data, the distribution of length gaps closely matches that of the original dataset (Figure 5a), indicating that the improvement in data quality is not due to increased response length. Although our filtering relies solely on internal information, observing the external reward margin distribution (Figure 5b) before and after filtering reveals that, for external rewards, the primarily removed samples are still ambiguous ones, and the distribution shifts slightly to the right.

**Comparative Overlap Analysis: Model-Level and Method-Level Perspectives.** At the method level (Figure 5c (bottom)), our approach shows high overlap with IM (67.4%) and SDPO (63.2%) due to their shared reliance on internal model signals for selecting high-reward-margin samples. Nevertheless, AlignDiff can still identify more representative and higher-quality data beyond the

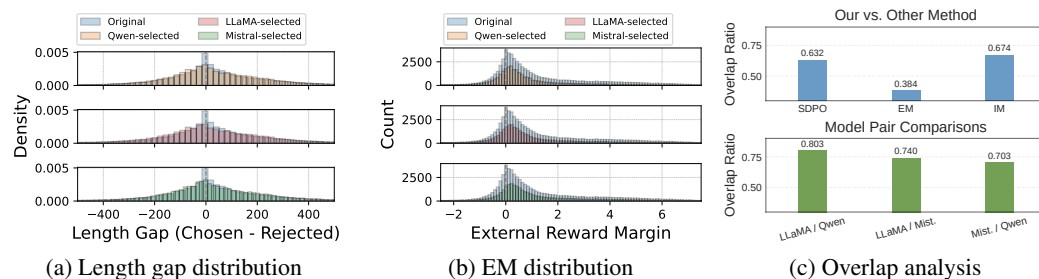

(a) Length gap distribution    (b) EM distribution    (c) Overlap analysis

Figure 5: *External Reward margin optimization with length gap preserved ((a) and (b)).* The filtered data caused almost no change in the length gap distribution, while optimizing the external reward margin distribution, reducing ambiguous samples, and shifting the overall distribution slightly to the right. Panel (c) shows the comparative analysis from method and model Perspectives.

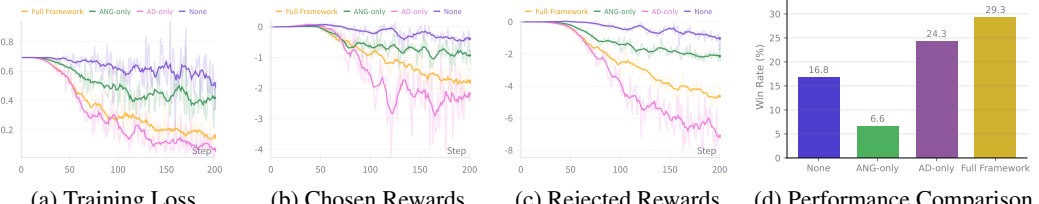

(a) Training Loss    (b) Chosen Rewards    (c) Rejected Rewards    (d) Performance Comparison

Figure 6: *Training Dynamics and Performance*. Dynamics of three metrics during DPO training: (a) training loss; (b) rewards of chosen responses; (c) rewards of rejected responses, each showing results for different filtering methods: None, AD-only, ANG-only, and the Full Framework; and (d) performance comparison of the methods. *Although AD-only achieves lower loss, it is constrained by the squeezing effect, limiting its performance, whereas ANG-only mitigates the squeezing effect but struggles to converge and learn properly, resulting in suboptimal performance.*

overlapping subset. In contrast, the overlap with EM is significantly lower (38.4%) because EM relies on external models for multi-dimensional fine-grained scoring, making its selection criteria fundamentally different from internal-signal-based methods. This highlights the distinctions and complementarity between the two types of methods. At the model level (Figure 5c (top)), the high overlap rates indicate consistency among different model architectures in identifying high-quality samples, possibly stemming from their shared capabilities in language understanding and data quality assessment, which also validates the robustness of our method across different architectures.

## 5.2 TRAINING DYNAMICS AND EFFECTIVENESS

We examine training stability, reward quality, and model performance. We find that (1) **alignment discrepancy provides cleaner optimization signals** (Figure 6a): After incorporating alignment differences, the full framework trains more stably than ANG-only, and converges to a level not far from that of training based on $R_{AD}$. Although the AD-only loss is lower, the final performance (Figure 6d) is not as good as that of the full framework, which further illustrates the impact of the squeezing effect. (2) **Mitigating the squeezing effect can be achieved by combining it with sample-difficulty-based selection** (Figure 6b and 6c): learning clearly preferred pairs, consisting of difficult chosen responses and easier rejected responses, stabilizes the implicit rewards of chosen and rejected responses, prevents the model from deviating excessively from the reference policy, and ensures relatively stable gradient signals. (3) **Combining AD and ANG in data selection can enhance performance** (Figure 6d): Without clear preferences, the model cannot learn effectively, resulting in worse performance than no filtering at all; without the stimulation of difficult samples, the model also cannot achieve better performance. (4) **Leading to greater preference consistency (Figure 7)**: We compute the AvgNLLs of chosen and rejected samples on the small, high-quality

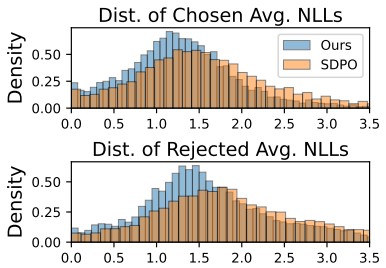

Figure 7: Dist. of Avg. NLLs. The model trained on our filtered data exhibits a tighter and left-shifted distribution of Avg. NLLs compared to SDPO.

preference dataset argilla/dpo-mix-7k[4] via forced decoding, and compare the results with those of the best-performing baseline SDPO, which relies solely on implicit rewards. The results show that the model trained on our filtered data exhibits a left-shifted and sharper average-NLL distribution for both chosen and rejected responses. This indicates that the model produces preferred responses more consistently. We further assess the impact of our filtered data on WPO (Zhou et al., 2024), the strongest DPO variant from the SDPO paper. LLaMA-3-8B-SFT trained on the original dataset achieves a win rate of 19.37%, whereas using our filtered data increases the win rate to 31.59%, highlighting the substantial improvement brought by AlignDiff.

## 5.3 ON THE UTILITY OF DIFFICULTY-BASED CURRICULUM LEARNING

In this work, we adopt the ANG as a metric to measure sample difficulty, where larger gaps indicate harder examples. Building on the idea of curriculum learning used in DPO (Pattnaik et al., 2024), we investigate whether a difficulty-based curriculum strategy can further

Table 2: Sample Ordering Strategies Comparison.

| Sample Ordering Strategy | LC(%) | WR(%) | Length |
|---|---|---|---|
| Random Order (baseline) | 33.38 | 33.81 | 2,035 |
| Hard-to-Easy | 27.72 | 27.95 | 2,005 |
| Easy-to-Hard | **36.40** | **36.89** | 2,064 |

improve model performance. We conduct an analytical experiment using the Qwen2.5-8B-SFT model, where the final 30k preference data selected by the whole AlignDiff framework is organized according to three difficulty-based sample ordering strategies: **(1) random order**, serving as a baseline without any curriculum structure; **(2) hard-to-easy order**, where more difficult samples are presented to the model first; and **(3) easy-to-hard order**, where simpler samples are introduced first to enhance the model's capability gradually. The results in Table 2 indicate significant performance differences under different training sample orders. For the random order, LC/WR are 33.38/33.81; for the hard-to-easy order, they drop to 27.72/27.95; and for the easy-to-hard order, performance is the best, with LC/WR increasing to 36.40/36.89. Therefore, the easy-to-hard training strategy can further improve model performance, and the training order has an important impact on the outcome.

## 5.4 COMPUTATIONAL EFFICIENCY COMPARISON

To evaluate the trade-off between computational cost incurred during the data selection and model performance, we compare Align-Diff with several baselines, including IM, EM, and SDPO. We report the GPU hours required for each method and compute efficiency as performance (average WR) per GPU hour. Please note that the computation of EM relies on the Qwen2.5-72B model for

Table 3: Efficiency Comparison across Methods.

| Method | GPU Hours | Avg. WR | Efficiency |
|---|---|---|---|
| IM | 43 | 24.6 | 0.57 |
| EM | 128 | 17.2 | 0.13 |
| SDPO | 162 | 25.6 | 0.16 |
| AlignDiff | 80.5 | 30.2 | 0.38 |

scoring across four dimensions, which leads to a relatively high computational cost. The detailed calculation of GPU hours for these methods can be found in the Appendix K. Table 3 shows that, compared with EM and SDPO, our method achieves a better balance between efficiency and performance. Although its efficiency is lower than IM's, our method attains superior performance while mitigating the squeeze effect.

## 6 CONCLUSION

We propose AlignDiff, a two-stage framework for filtering preference data using only the model's internal signals. Our approach effectively removes ambiguous or conflicting samples by leveraging both positive and inverse preference signals through bidirectional alignment discrepancies. We further refine data selection using the average negative log-likelihood gap to identify high-quality, informative pairs. Both stages depend entirely on the model's internal evaluation signals, avoiding potential distributional shifts caused by external reward functions and ensuring that the selected data aligns closely with the model's inherent learning tendencies. Experiments demonstrate that our method substantially improves alignment across different base LLMs. On AlpacaEval 2.0, it surpasses existing preference data selection baselines, achieving a 3.9% gain in length-controlled performance while maintaining comparable response lengths to SDPO. Incorporating difficulty-based curriculum learning provides additional improvements, highlighting the effectiveness of our signal-driven data selection strategy.

---

[4]https://huggingface.co/datasets/argilla/dpo-mix-7k

ETHICS STATEMENT

This work investigates preference data filtering for aligning LLMs with human preferences. All experiments are conducted on publicly available datasets, without the use of sensitive personal information or human subject studies. While AlignDiff aims to improve the robustness and efficiency of alignment by filtering high-quality preference data, we acknowledge that preference datasets may still contain biases, inconsistencies, or culturally specific values that could influence model behavior. We encourage future research to further examine fairness, cultural diversity, and inclusivity in alignment datasets.

REPRODUCIBILITY STATEMENT

All models and datasets used in this work are publicly available and open-source. Detailed training procedures can be found in Appendix §G, and specific evaluation methods are provided in Appendix §F. In addition, to enable readers to reproduce the experimental results reported in this paper, we have made our code publicly available at https://anonymous.4open.science/r/AlignDiff-3311.

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

# A    LIMITATION

AlignDiff relies on intrinsic model signals (e.g., likelihood), which may vary with model scale and architecture, so it may be necessary to adjust the corresponding hyperparameters for different models. Moreover, although AlignDiff demonstrates promising improvements, the filtering process inevitably discards part of the training data, which may also exclude some valuable but subtle preference signals. These aspects constitute important directions for future work.

# B    STATEMENT ON THE USE OF LLMS

In the preparation of this work, LLMs were employed solely for language refinement and stylistic improvements. All innovative content in the work was generated by the authors without assistance from LLMs. The use of LLMs was limited to enhancing clarity, grammar, and readability, ensuring that the manuscript communicates the authors' original work effectively and accurately.

# C    PSEUDOCODE FOR ALIGNDIFF

---

**Algorithm 1 AlignDiff**

---

**Input:** preference dataset $\mathcal{D}$ of triples $(\mathbf{x}, \mathbf{y}_w, \mathbf{y}_l)$; reference sft model $\pi_{\text{ref}}$; alignment discrepancy threshold $\tau > 0$; ANG filtering percentile $K$.

**# Stage 1: Alignment Discrepancy Filtering**
 Train $\pi_{\boldsymbol{\theta}}^{\text{positive}}$ and $\pi_{\boldsymbol{\theta}}^{\text{inverse}}$ on $\mathcal{D}$ using $\mathcal{L}_{\text{DPO}}^{+}$ and $\mathcal{L}_{\text{DPO}}^{-}$ respectively.   ▷ Using Equations 6, 7.
 Initialize an intermediate dataset $\tilde{\mathcal{D}} \leftarrow \emptyset$.
**for** each preference pair $(\mathbf{x}, \mathbf{y}_w, \mathbf{y}_l) \in \mathcal{D}$ **do**
  Compute $R_{\text{AD}}(\mathbf{y}_w, \mathbf{y}_l | \mathbf{x})$ using $\pi_{\boldsymbol{\theta}}^{\text{positive}}$ and $\pi_{\boldsymbol{\theta}}^{\text{inverse}}$.   ▷ Using Equation. 8
 **if** $R_{\text{AD}}(\mathbf{y}_w, \mathbf{y}_l | \mathbf{x}) > \tau$ **then**
   Add $(\mathbf{x}, \mathbf{y}_w, \mathbf{y}_l)$ to $\tilde{\mathcal{D}}$.   ▷ Clear positive preference
 **else if** $R_{\text{AD}}(\mathbf{y}_w, \mathbf{y}_l | \mathbf{x}) < -\tau$ **then**
   Add $(\mathbf{x}, \mathbf{y}_l, \mathbf{y}_w)$ to $\tilde{\mathcal{D}}$.   ▷ Clear inverse preference, swap labels
 **end if**   ▷ Ambiguous samples where $|R_{\text{AD}}| \leq \tau$ are discarded.
**end for**

**# Stage 2: Sample Difficulty-aware Calibration**
 Initialize a list for scored samples $\mathcal{S} \leftarrow [\,]$.
**for** each sample $(\mathbf{x}, \tilde{\mathbf{y}}_w, \tilde{\mathbf{y}}_l) \in \tilde{\mathcal{D}}$ **do**
  Compute $\text{ANG}(\tilde{\mathbf{y}}_w, \tilde{\mathbf{y}}_l)$ using the reference model $\pi_{\text{ref}}$.
  Add the tuple $(\mathbf{x}, \tilde{\mathbf{y}}_w, \tilde{\mathbf{y}}_l, \text{ANG}(\tilde{\mathbf{y}}_w, \tilde{\mathbf{y}}_l))$ to $\mathcal{S}$.
**end for**
 Sort $\mathcal{S}$ in descending order based on the ANG.
 Let $\mathcal{S}_{\text{top-K}}$ be the subset containing the top $K$ percentile of samples from $\mathcal{S}$.
 Initialize the final dataset $\mathcal{D}_{\text{final}} \leftarrow \emptyset$.
**for** each sample $(\mathbf{x}, \tilde{\mathbf{y}}_w, \tilde{\mathbf{y}}_l, \text{ANG}(\tilde{\mathbf{y}}_w, \tilde{\mathbf{y}}_l)) \in \mathcal{S}_{\text{top-K}}$ **do**
  Add $(\mathbf{x}, \tilde{\mathbf{y}}_w, \tilde{\mathbf{y}}_l)$ to $\mathcal{D}_{\text{final}}$.
**end for**
**Output:**
 The final high-quality filtered dataset $\mathcal{D}_{\text{final}}$.

---

# D    RELATED WORK

## D.1    PREFERENCE LEARNING FOR ALIGNMENT

Early work (Peng et al., 2023; Ji et al., 2024; Bai et al., 2023) predominantly employs the RLHF (Ouyang et al., 2022) framework, integrating algorithms such as PPO (Schulman et al.,

2017) with explicit reward modeling to improve alignment between models and human preferences. DPO (Rafailov et al., 2023) has been proposed as an efficient alternative approach that leverages implicit reward signals to directly optimize models using preference data, thereby streamlining the training process and substantially reducing computational costs. Based on this foundation, more advanced preference learning algorithms such as IPO (Azar et al., 2023), KTO (Ethayarajh et al., 2024), and SimPO (Meng et al., 2024) have emerged, aiming to improve the efficiency of preference modeling and enhance the alignment performance of language models. However, these algorithms all rely on high-quality preference data, and ensuring the quality of such data has become one of the key bottlenecks in further improving alignment performance.

### D.2 DATA SELECTION IN PREFERENCE OPTIMIZATION

The alignment performance of a large language model is largely dependent on the quality of the preference data. Early preference optimization work (Touvron et al., 2023; Cui et al., 2023; Wang et al., 2023b; Grattafiori et al., 2024) adopts data selection techniques inherited from the pre-training and instruction-tuning stages, such as deduplication, quality classifiers, and heuristic filtering. These approaches typically rely on an external reward signal, which refers to the scores assigned by a reward model or an LLM-as-a-judge to evaluate candidate responses, and apply rejection sampling to construct preference pairs. However, recent studies (Wu et al., 2024; Khaki et al., 2024) suggest that, compared to methods that evaluate and filter individual responses based on absolute quality, selecting response pairs according to their reward gap is more beneficial for preference optimization, as it provides stronger training signals for alignment. Inspired by this, Morimura et al. (2024); Hu et al. (2024b) leveraged the external reward gap to filter preference data. Unlike using external reward signals, Kim et al. (2025) trained a weakly aligned model to compute implicit rewards and use the implicit reward gap (derived from the logit differences in the DPO objective and reflecting the model's internal preferences) as the basis for data selection. Deng et al. (2025) found that the external reward gap and the implicit reward gap exhibit a notably weak correlation; thus, they combined both the external reward gap and the implicit reward gap to refine the preference pair selection process. Overall, this phenomenon reflects a paradigm shift in preference data selection, gradually transitioning from reliance on external reward signals to reliance on internal ones.

## E HYPERPARAMETER STUDY ON THE ALIGNMENT DISCREPANCY THRESHOLD OF THE MISTRAL MODEL

In this section, we present a hyperparameter study on the alignment discrepancy (AD) threshold $\tau$ for the Mistral model. Since the AD values computed by different models vary, it is necessary to analyze the threshold separately for each model. To balance the threshold $\tau$ and the corresponding data volume, we select three $\tau$ values: 60, 80, and 100 for the Mistral model. The reason for this choice is that when $\tau$ is set too small (e.g., $\tau = 20$), the amount of data filtered out is too limited, resulting in negligible differences in training performance. We aim for each $\tau$ to correspond to about 5k data points to ensure the effectiveness of the analysis. As shown in Figure 8, setting $\tau = 80$ achieves an optimal trade-off between data quality and quantity for Mistal.

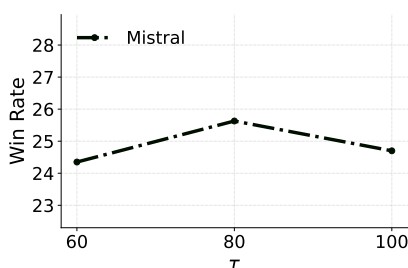

Figure 8: Hyperparameter Study $\tau$ for the Mistral Model.

## F EVALUATION SETTINGS

### F.1 EVALUATION DATASETS

**AlpacaEval 2.0**   *AlpacaEval 2.0*[5] is an automated evaluation tool designed to efficiently and cost-effectively assess the performance of instruction-following language models. It is based on the

---

[5]https://github.com/tatsu-lab/alpaca_eval

AlpacaFarm dataset and focuses on testing the models' ability to understand and execute general user instructions. Specifically, we conduct pairwise comparisons on 805 test examples, where the outputs of the DPO-trained model are compared against those of the strong baseline *GPT-4-1106-Preview*. The preferences are judged by the automatic evaluator *deepseek-v3-03-24* (See Appendix F.2 for the justification of using this as the evaluator). We utilize a fixed decoding temperature (T = 0.9) for all model generation in the experiments. To align with the training phase, we also limit the total length of input and generated tokens to 2048 (max_length=2048) during inference.

**MT-Bench**  *MT-Bench*[6] is a benchmark framework for evaluating the performance of large language models (LLMs) in multi-turn dialogues, designed to address the limitations of traditional evaluations (such as MMLU and HELM) in open-ended tasks and human preference alignment. We conduct the evaluation on FastChat, using *GPT-4 Turbo* as the judge model.

### F.2 Justification for Using DeepSeek-V3 as the Evalueator

Due to the extensive experiments conducted in this paper, the use of gpt4_turbo in the official *AlpacaEval 2.0* implementation is expensive for us. Therefore, we opted for the more powerful and cost-effective model *deepseek-v3-0324*. We analyzed our annotator using the analyze_evaluators command in *AlpacaEval 2.0* and compared it with the official annotator in Table 4. The results show that our annotator outperforms the official one in human consistency, Spearman correlation, and Pearson correlation, while being significantly more cost-effective.

Table 4: Comparison of AlpacaEval 2.0 Annotators. The better results under each metric in the table are highlighted in **bold**.

| Annotator | Human Agreement | Price | Spearman Corr. | Pearson Corr. | Bias | Variance |
|---|---|---|---|---|---|---|
| DeepSeek-V3 | **67.27** | **0.12** | **0.95** | **0.87** | **32.19** | **16.45** |
| GPT-4 Turbo | 65.73 | 4.32 | 0.78 | 0.77 | 33.90 | 23.65 |

## G  Details of DPO Training

All DPO experiments are conducted on $8\times$ L40 GPUs. Specifically, we use the AdamW optimizer with a cosine learning rate scheduler and a learning rate of 5e-7; a warmup ratio of 10% is applied at the beginning of training. All models are only trained for 1 epoch over the training set. For the hyper-parameter $\beta$ of DPO, we use a fixed value of $\beta = 0.01$. Input sequences are truncated or padded to a maximum length of 2048 tokens.

## H  Details of Baselines

**External Reward Margin (EM).**   As a commonly used filtering strategy, we select four key evaluation dimensions (helpfulness, instruction following, honesty, and truthfulness) and leverage the powerful Qwen2.5-72B-Instruct model to score the chosen and rejected responses for each preference pair. Unlike the conventional approach of assigning discrete integer scores, we follow the method proposed in Lai et al. (2025), performing probability-weighted aggregation over score tokens to obtain more fine-grained scores for each dimension. Details of the prompt template can be found in Table 6. Specifically, we prompt the model to assign a score from 1 to 9 for each dimension, denoted as $s \in \{1, 2, \ldots, 9\}$. The score for each dimension is computed as:

$$\text{Score} = \sum_{s=1}^{9} s \times P(s), \tag{13}$$

where $P(s)$ denotes the probability assigned by the model to the score token s, typically obtained via softmax. This formulation allows for a smoother and more fine-grained score. The final score for each sample pair is then computed as the average of the scores across the four dimensions. We then

---

[6]https://github.com/lm-sys/FastChat/tree/main/fastchat/llm_judge#mt-bench

consider the samples with the largest score difference between chosen and rejected as high-quality preference examples for filtering.

**PPLGAP.** To identify high-quality preference pairs, we use PPL Gap as a difficulty-based filtering metric. Given a preference pair $(\mathbf{y}_w, \mathbf{y}_l)$, we define:

$$\mathrm{PPLGap}(\mathbf{y}_w, \mathbf{y}_l) = \mathrm{PPL}(\mathbf{y}_w) - \mathrm{PPL}(\mathbf{y}_l), \tag{14}$$

where PPL is computed as:

$$\mathrm{PPL}(\mathbf{y}) = \exp\left(\frac{1}{|\mathbf{y}|}\sum_{t=1}^{|\mathbf{y}|} -\log P(y_t \mid y_{<t})\right). \tag{15}$$

A larger PPLGap indicates the chosen response is much more likely under the model than the rejected one, suggesting a clearer preference. We select preference pairs with the highest PPLGap values as high-quality data.

**Implicit reward margin (IM).** We use the corresponding SFT model as the reference model and train the policy model on the entire UltraFeedback_Binarized dataset. Then, we compute the implicit reward margin for all data in this dataset using these three SFT models, as defined in Equation 5. Samples with the highest implicit reward margins are regarded as high-quality preference examples, as they better reflect a strong alignment between the model and human preferences.

**External & Implicit reward margin (IM&EM).** We aggregate the external reward margin $M_{ex}$ and the implicit reward margin $M_{im}$ following the method described in Deng et al. (2025). Specifically, we first transform the margin values into margin-guided probabilities through a simple linear transformation:

$$\mathbb{P}(M) = \frac{\mathrm{clip}(M, M_1, M_2) - M_1}{M_2 - M_1}, \quad \text{for} \quad M \in \{M_{ex}, M_{im}\}, \tag{16}$$

where $\mathrm{clip}(M) = min(max(M, M_1), M_2))$ and $(M_1, M_2)$ are tuning parameters. As provided in the original paper, we adopt the optimal settings for $M_1$ and $M_2$. Consequently, we obtain:

$$P(\mathbf{y}_w \geq \mathbf{y}_l | M_{ex}, M_{im}) = \frac{P(M_{ex})P(M_{im})}{P(M_{ex})P(M_{im}) + (1 - P(M_{ex})) \cdot (1 - P(M_{im}))}. \tag{17}$$

We then select the samples with the highest probabilities obtained above as high-quality preference examples.

**R.I.P.** Yu et al. (2025) proposed that, under the condition of maintaining a positive external reward margin, preference pairs in which the rejected response exceeds a certain length threshold are more likely to be high-quality. Therefore, following the setup in the original paper, we set the external reward margin threshold to 0.126 and select those samples with the longest rejected responses as high-quality preference data.

**SelectiveDPO.** Hu et al. (2024b) proposed that, by training six reference models to compute validation loss for identifying sample difficulty, then filtering out overly difficult samples that exceed the model's capacity, and conducting preference alignment training only on easy samples within the model's capacity. The validation loss $\mathrm{VL}(\mathbf{y}_w, \mathbf{y}_l | \mathbf{x})$ can be computed as:

$$\mathrm{VL}(\mathbf{y}_w, \mathbf{y}_l | \mathbf{x}) = -\log \sigma\left(\beta \log \frac{\pi_{\boldsymbol{\theta}}(\mathbf{y}_w | \mathbf{x})}{\pi_{\mathrm{ref}}(\mathbf{y}_w | \mathbf{x})} - \beta \log \frac{\pi_{\boldsymbol{\theta}}(\mathbf{y}_l | \mathbf{x})}{\pi_{\mathrm{ref}}(\mathbf{y}_l | \mathbf{x})}\right). \tag{18}$$

**In essence, the validation loss $\mathrm{VL}(\mathbf{y}_w, \mathbf{y}_l | \mathbf{x})$ is equivalent to the implicit reward margin. This implies that the data selected by their method can also be filtered using implicit reward margins derived from six different models.** We directly use the top 50% examples selected from UltraFeedback_Binarized as identified by the paper, which represents the best-performing setting.

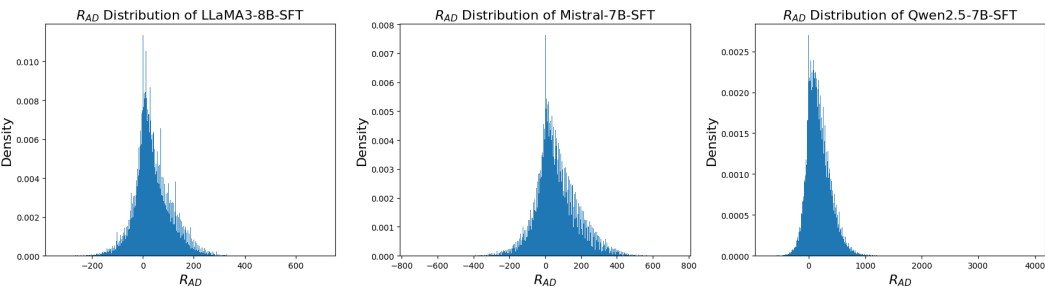

Figure 9: Distribution of $R_{AD}$ for three SFT models on the UltraFeedback dataset.

# I MATHEMATICAL DERIVATIONS

## I.1 THE DETAILED DERIVATION OF ALIGNMENT DISCREPANCY

The Alignment Discrepancy $R_{\mathrm{AD}}(\mathbf{y}_w, \mathbf{y}_l|\mathbf{x})$ is defined based on the difference in implicit reward margins between $\pi_{\boldsymbol{\theta}}^{\mathrm{positive}}$ and $\pi_{\boldsymbol{\theta}}^{\mathrm{inverse}}$, with the formula:

$$R_{\mathrm{AD}}(\mathbf{y}_w, \mathbf{y}_l|\mathbf{x}) = \mathrm{M}_{\mathrm{im}}^{\mathrm{positive}}(\mathbf{y}_w, \mathbf{y}_l|\mathbf{x}) - \mathrm{M}_{\mathrm{im}}^{\mathrm{inverse}}(\mathbf{y}_w, \mathbf{y}_l|\mathbf{x}).$$

Based on the calculation formula for the implicit reward margin as shown in Equation 5, we can substitute and simplify it to obtain the final form:

$$
\begin{aligned}
R_{\mathrm{AD}}(\mathbf{y}_w, \mathbf{y}_l|\mathbf{x}) &= \mathrm{M}_{\mathrm{im}}^{\mathrm{positive}}(\mathbf{y}_w, \mathbf{y}_l|\mathbf{x}) - \mathrm{M}_{\mathrm{im}}^{\mathrm{inverse}}(\mathbf{y}_w, \mathbf{y}_l|\mathbf{x}) \\
&= \left( r_{\mathrm{implicit}}^{\mathrm{positive}}(\mathbf{y}_w|\mathbf{x}) - r_{\mathrm{implicit}}^{\mathrm{positive}}(\mathbf{y}_l|\mathbf{x}) \right) \\
&\quad - \left( r_{\mathrm{implicit}}^{\mathrm{inverse}}(\mathbf{y}_w|\mathbf{x}) - r_{\mathrm{implicit}}^{\mathrm{inverse}}(\mathbf{y}_l|\mathbf{x}) \right) \\
&= \left( \log \frac{\pi_{\boldsymbol{\theta}}^{\mathrm{positive}}(\mathbf{y}_w|\mathbf{x})}{\pi_{\mathrm{ref}}(\mathbf{y}_w|\mathbf{x})} - \log \frac{\pi_{\boldsymbol{\theta}}^{\mathrm{positive}}(\mathbf{y}_l|\mathbf{x})}{\pi_{\mathrm{ref}}(\mathbf{y}_l|\mathbf{x})} \right) \\
&\quad - \left( \log \frac{\pi_{\boldsymbol{\theta}}^{\mathrm{inverse}}(\mathbf{y}_w|\mathbf{x})}{\pi_{\mathrm{ref}}(\mathbf{y}_w|\mathbf{x})} - \log \frac{\pi_{\boldsymbol{\theta}}^{\mathrm{inverse}}(\mathbf{y}_l|\mathbf{x})}{\pi_{\mathrm{ref}}(\mathbf{y}_l|\mathbf{x})} \right) \\
&= \left( \log \frac{\pi_{\boldsymbol{\theta}}^{\mathrm{positive}}(\mathbf{y}_w|\mathbf{x})}{\pi_{\boldsymbol{\theta}}^{\mathrm{positive}}(\mathbf{y}_l|\mathbf{x})} - \log \frac{\pi_{\mathrm{ref}}(\mathbf{y}_w|\mathbf{x})}{\pi_{\mathrm{ref}}(\mathbf{y}_l|\mathbf{x})} \right) \\
&\quad - \left( \log \frac{\pi_{\boldsymbol{\theta}}^{\mathrm{inverse}}(\mathbf{y}_w|\mathbf{x})}{\pi_{\boldsymbol{\theta}}^{\mathrm{inverse}}(\mathbf{y}_l|\mathbf{x})} - \log \frac{\pi_{\mathrm{ref}}(\mathbf{y}_w|\mathbf{x})}{\pi_{\mathrm{ref}}(\mathbf{y}_l|\mathbf{x})} \right) \\
&= \log \frac{\pi_{\boldsymbol{\theta}}^{\mathrm{positive}}(\mathbf{y}_w|\mathbf{x})}{\pi_{\boldsymbol{\theta}}^{\mathrm{positive}}(\mathbf{y}_l|\mathbf{x})} - \log \frac{\pi_{\boldsymbol{\theta}}^{\mathrm{inverse}}(\mathbf{y}_w|\mathbf{x})}{\pi_{\boldsymbol{\theta}}^{\mathrm{inverse}}(\mathbf{y}_l|\mathbf{x})}.
\end{aligned}
$$

# J ADDITIONAL EXPERIMENTAL RESULTS

To verify whether the effectiveness of this method is not limited to AlpacaEval 2.0 / MT-Bench, we extended the evaluation to more diverse data distributions. We assessed the performance of this method and other baselines on GPQA Rein et al. (2023), Toxigen Hartvigsen et al. (2022), TruthfulQA Lin et al. (2022), MMLU Hendrycks et al. (2021), and Winogrande Sakaguchi et al. (2019), with the results shown in Table 5. It can be seen that the differences between the methods are not significant. We also show the distribution of $R_{AD}$ for three SFT models on the UltraFeedback dataset in Figure 9.

Table 5: *Performance Comparison on Multi-Task Benchmarks (GPQA / Toxigen / TruthfulQA / MMLU / Winogrande) Using DPO-Trained Models with Various Data Subsets*. The base models are LLaMA-3-8B-SFT, Mistral-7B-SFT, and Qwen2.5-7B-SFT.

| Method | GPQA | Toxigen | TruthfulQA | MMLU | Winogrande | AVG |
|---|---|---|---|---|---|---|
| LLaMA-3-8B-SFT | | | | | | |
| PPLGAP | 0.328 | 0.427 | 0.493 | 0.613 | 0.729 | 0.518 |
| EM | 0.313 | 0.430 | 0.521 | 0.624 | 0.719 | 0.521 |
| IM | 0.317 | 0.429 | 0.488 | 0.622 | 0.719 | 0.515 |
| IM & Em | 0.308 | 0.428 | 0.490 | 0.623 | 0.717 | 0.513 |
| LCPP | 0.332 | 0.427 | 0.498 | 0.612 | 0.730 | 0.520 |
| R.I.P | 0.324 | 0.427 | 0.492 | 0.620 | 0.721 | 0.517 |
| SDPO | 0.319 | 0.430 | 0.485 | 0.620 | 0.717 | 0.514 |
| AlignDiff | 0.335 | 0.428 | 0.493 | 0.621 | 0.722 | 0.520 |
| Mistral-7B-SFT | | | | | | |
| PPLGAP | 0.319 | 0.559 | 0.528 | 0.571 | 0.750 | 0.545 |
| EM | 0.304 | 0.587 | 0.619 | 0.584 | 0.738 | 0.566 |
| IM | 0.304 | 0.579 | 0.620 | 0.571 | 0.729 | 0.560 |
| IM & Em | 0.306 | 0.572 | 0.635 | 0.566 | 0.741 | 0.564 |
| LCPP | 0.308 | 0.476 | 0.496 | 0.583 | 0.740 | 0.520 |
| R.I.P | 0.315 | 0.577 | 0.577 | 0.582 | 0.739 | 0.558 |
| SDPO | 0.301 | 0.578 | 0.562 | 0.578 | 0.732 | 0.550 |
| AlignDiff | 0.317 | 0.572 | 0.586 | 0.582 | 0.740 | 0.559 |
| Qwen2.5-7B-SFT | | | | | | |
| PPLGAP | 0.339 | 0.567 | 0.611 | 0.710 | 0.720 | 0.589 |
| EM | 0.346 | 0.569 | 0.615 | 0.709 | 0.710 | 0.590 |
| IM | 0.344 | 0.569 | 0.603 | 0.709 | 0.707 | 0.586 |
| IM & Em | 0.344 | 0.569 | 0.603 | 0.710 | 0.706 | 0.586 |
| LCPP | 0.326 | 0.569 | 0.602 | 0.709 | 0.707 | 0.583 |
| R.I.P | 0.348 | 0.568 | 0.608 | 0.709 | 0.707 | 0.588 |
| SDPO | 0.344 | 0.569 | 0.603 | 0.709 | 0.710 | 0.587 |
| AlignDiff | 0.342 | 0.569 | 0.609 | 0.709 | 0.708 | 0.587 |

## K  CALCULATION OF GPU HOURS FOR THE METHODS

In our experiments, since both training and inference were consistently conducted on 8×L40 GPUs, it is straightforward to estimate the computational cost of each method. For the four main methods compared in this paper, the estimates are as follows:

- **EM**: This method requires using the Qwen2.5-72B-Instruct model to score along four dimensions. Deploying this model requires 4 L40 GPUs, and the inference for a single response takes approximately 4 hours. Since each prompt corresponds to two responses (chosen and rejected), the estimated computational cost of EM is: $4 \times 8 \times 4 = 128$ GPUh.

- **IM**: This method requires training a forward DPO model on the original dataset, which takes about 8 GPUs × 4h = 32 GPUh. Then, computing the implicit reward margin takes about 11 GPUh. Thus, the total cost of IM is: $32 + 11 = 43$ GPUh.

- **AlignDiff**: This method requires training a forward DPO model on the original dataset and a reverse DPO model on the inverted dataset, taking about 8 × 4 × 2 = 64 GPUh in total. In addition, computing the forward and reverse implicit reward margins and the NLL of the original model takes about 11 + 5.5 = 16.5 GPUh. Therefore, the total cost of AlignDiff is: $64 + 16.5 = 80.5$ GPUh.

- **SDPO**: This method requires training 6 reference models on half of the original dataset, which costs about 6 × 8 × 2 = 96 GPUh. Then, each model requires implicit reward margin computation, costing about 6 × 11 = 66 GPUh. Thus, the total cost of SDPO is: $96 + 66 = 162$ GPUh.

## L  PROMPT TEMPLATE

In this section, we present all the prompt templates used in this paper. To evaluate preference pairs effectively, we design a structured prompt template that guides the Qwen2.5-72B-Instruct model

---

**Prompt Template for LLM-as-a-judge Evaluation**

You are an evaluator tasked with assigning a score from 1 to 9 based on the criteria below, to assess the quality of a specific aspect of the data entry.

### Evaluation Task:

Assess the given data entry based on the following aspect: **{evaluation_aspect}**. For this task, your goal is to evaluate how well the **{evaluation_aspect}** is demonstrated in the provided **Instruction** and **Response**. Be sure to consider the following:

- **{evaluation_aspect}** refers to a specific quality or characteristic in the data entry.
- Consider both the **Instruction** (the directive given) and the **Response** (the output provided in answer to the instruction).
- The evaluation should be based on how clearly, accurately, or relevantly the task is fulfilled according to the criteria listed below.
- Only assess the given **Instruction** and **Response**; do not consider any additional context or external information.

### Definition of {evaluation_aspect}:

- **{evaluation_aspect}** is {Definition}

### Scoring Criteria:

- **Score 1**: {criteria_for_1}
- **Score 2**: {criteria_for_2}
- **Score 3**: {criteria_for_3}
- **Score 4**: {criteria_for_4}
- **Score 5**: {criteria_for_5}
- **Score 6**: {criteria_for_6}
- **Score 7**: {criteria_for_7}
- **Score 8**: {criteria_for_8}
- **Score 9**: {criteria_for_9}

Each score corresponds to a specific level of quality, from 1 (lowest) to 9 (highest). Use the provided criteria to select the appropriate score.

### Data Entry to Evaluate:

- **Instruction**:
{instruction}

- **Response**:
{output}

### Response Format:

- Provide a **single score** from 1 to 9 in format of "score: <the score you give>" based on your evaluation of the **Instruction** and **Response**.

---

Figure 10: **Prompt Template for Scoring Preference Pairs Using Qwen2.5-72B-Instruct**.

in assigning scores across multiple evaluation dimensions. The specific design of our template is illustrated in Figure 10, which provides a clear example of how questions, responses, and scoring instructions are presented to the model. Specifically, we focus on four key dimensions: helpfulness, instruction following, honesty, and truthfulness. The detailed definitions of different dimensions and their specific scoring standards are presented in Table 6.

Table 6: Definitions of different dimensions and their specific scoring standards, with each dimension scored on a scale ranging from 1 to 9, each score corresponds to a specific standard.

| Dimension-1 & Definition | **Answer Accuracy:** Evaluate whether the response accurately addresses the instruction and completely fulfills the task. |
|---|---|
| **Scoring Standards** | 1: Completely incorrect, irrelevant to the instruction. |
| | 2: Partially correct, major omissions or errors in fulfilling the instruction. |
| | 3: Contains significant errors, unable to fully address the core task. |
| | 4: Partially correct, missing key details or addressing the wrong aspect of the instruction. |
| | 5: Mostly accurate, but contains some errors or omissions. |
| | 6: Mostly correct, though missing small details or has minor inaccuracies. |
| | 7: Largely accurate and complete, but may lack small details or have minimal errors. |
| | 8: Fully accurate, completely addresses the instruction, minimal flaws. |
| | 9: Perfectly accurate, fully aligns with the instruction, no omissions. |
| Dimension-2 & Definition | **Helpfulness:** Assess whether the response provides valuable and actionable information that directly aids the user in achieving their objective. |
| **Scoring Standards** | 1: Response is entirely unhelpful and irrelevant, offering no value. |
| | 2: Response is mostly unhelpful, with very little relevance to the user's needs. |
| | 3: Provides limited help, but the response is vague, incomplete, or only partially relevant. |
| | 4: Response is somewhat helpful, but important details or steps are missing. |
| | 5: Moderately helpful, addressing the main query but lacking depth or precision. |
| | 6: Mostly helpful and clear, with minor gaps or room for further detail. |
| | 7: Helpful and mostly comprehensive, fulfilling the majority of the user's request. |
| | 8: Highly helpful and clear, with only minimal areas for potential improvement. |
| | 9: Exceptionally helpful, fully addressing the user's needs in a complete and actionable manner. |
| Dimension-3 & Definition | **Honesty:** Evaluate whether the response is truthful and avoids misleading or fabricating information. |
| **Scoring Standards** | 1: Response is entirely dishonest, fabricated, or intentionally misleading. |
| | 2: Response includes significant falsehoods or misleading information. |
| | 3: Some information is correct, but key elements are false or deceptive. |
| | 4: Mostly truthful, but contains noticeable inaccuracies or vague claims. |
| | 5: Generally honest, though minor factual errors or omissions exist. |
| | 6: Honest and clear, with very few small inaccuracies or overly cautious phrasing. |
| | 7: Mostly accurate and honest, with no significant misleading elements. |
| | 8: Entirely honest and accurate, only minor potential for clarification needed. |
| | 9: Flawlessly honest, all information is accurate and presented transparently. |
| Dimension-4 & Definition | **Instruction Following:** Determine if the response adheres precisely to the instructions provided, fulfilling the request as intended. |

Table 6: Definitions of different dimensions and their specific scoring standards, with each dimension scored on a scale ranging from 1 to 9, each score corresponds to a specific standard. (Continued)

| | |
|---|---|
| **Scoring Standards** | 1: Completely disregards the instruction, with no relation to the request. |
| | 2: Fails to follow the instruction significantly, with only minor relevant elements. |
| | 3: Partially follows the instruction, but with major omissions or errors. |
| | 4: Somewhat follows the instruction, though important aspects are overlooked. |
| | 5: Follows the instruction moderately well, with noticeable gaps or misinterpretations. |
| | 6: Mostly adheres to the instruction, with minor deviations or missed nuances. |
| | 7: Adheres to the instruction well, with only slight areas for improvement. |
| | 8: Accurately follows the instruction, with minimal need for refinement. |
| | 9: Perfectly adheres to the instruction, fulfilling every aspect flawlessly. |
| **Dimension-5 & Definition** | **Truthfulness:** Assess whether the response is factually accurate and based on verified knowledge or reasoning. |
| **Scoring Standards** | 1: Response is completely false, with no factual basis or accuracy. |
| | 2: Response is mostly false, with very few correct facts. |
| | 3: Response has a mix of true and false information, with major inaccuracies. |
| | 4: Response is somewhat accurate but includes noticeable factual errors. |
| | 5: Response is generally accurate but contains some minor factual inaccuracies. |
| | 6: Mostly truthful and fact-based, with minimal errors or ambiguities. |
| | 7: Highly accurate and truthful, with no significant errors or misleading information. |
| | 8: Entirely truthful and accurate, with only trivial areas for clarification or nuance. |
| | 9: Flawlessly truthful, presenting facts with utmost accuracy and precision. |