# OpenReview forum: "AlignDiff: Exploiting Model-Intrinsic Information for Better Preference Data Selection"
_ICLR.cc/2026/Conference — Submitted to ICLR 2026_

### Official Review · Reviewer_D9co · 2025-10-25

**Soundness:** 3
**Presentation:** 3
**Contribution:** 3
**Rating:** 6
**Confidence:** 2

**Summary:**

The paper introduces AlignDiff, a two-stage, intrinsic-signal preference data selector for DPO:

Alignment Discrepancy (RAD) combines positive and inverse implicit reward margins to keep clear-preference pairs, flip clearly inverted ones, and drop ambiguous ones.

Difficulty calibration (ANG) favors hard chosen responses (and relatively easy rejected responses) to avoid the DPO squeezing effect.

**Strengths:**

Positive vs inverse signals are complementary; RAD is an elegant, model-intrinsic indicator of preference consistency.

Strong empirical with cross-model gains; preserves length-gap distribution while shifting external reward margin right; solid ablations & training-dynamics analyses.

**Weaknesses:**

Current results focus on UltraFeedback → AlpacaEval 2.0 / MT-Bench. Please expand to diverse distributions—e.g., multilingual prompts (zh/es), safety red-team splits, and domain-shift (OOD) sets. Add a human evaluation slice (stratified 200–500 items, double-annotated) to validate that gains aren’t artifacts of automatic metrics and to reduce the risk of metric gaming.

The naive IM&EM combination underperforms IM. Consider an aware fusion that (i) calibrates score scales, (ii) debiases for response length, and (iii) trains a lightweight selector (e.g., logistic/GBM) over both features.

Add multilingual, safety-critical, and OOD stress suites, and report human preference subsamples.

**Questions:**

see weaknesses

---

> ### Author Response · Authors · 2025-11-19
> **Response (Part 1 of 1)**
>
> Thank you for your thoughtful feedback and constructive suggestions! We have carefully revised the manuscript in accordance with your suggestions. Our key responses are summarized below:
>
> > Q1: Current results focus on UltraFeedback → AlpacaEval 2.0 / MT-Bench. Please expand to diverse distributions—e.g., multilingual prompts (zh/es), safety red-team splits, and domain-shift (OOD) sets. Add a human evaluation slice (stratified 200–500 items, double-annotated) to validate that gains aren’t artifacts of automatic metrics and to reduce the risk of metric gaming.
>
> **A1**:
>
>   - Thank you for the valuable suggestion. We agree that evaluating on more diverse distributions could further strengthen the evidence of our method’s generalization capability. We conducted additional evaluations on GPQA, Toxigen, TruthfulQA, MMLU, and Winogrande, with the detailed results shown in Table 5 of the paper. The results show that AlignDiff also nearly outperforms SDPO, the baseline that demonstrated the strongest performance in the main experiments.
>
> > Q2: The naive IM&EM combination underperforms IM. Consider an aware fusion that (i) calibrates score scales, (ii) debiases for response length, and (iii) trains a lightweight selector (e.g., logistic/GBM) over both features.
>
> **A2**:
>
>   - **IM and EM serve only as baselines and are not methods proposed by us; therefore, it may not be appropriate to regard them as a weakness of our approach**. We have followed the original papers’ settings and performed the corresponding score scaling calibration; however, such calibration seems infeasible for methods (ii) and (iii).
>
> > Q3: Add a human evaluation slice (stratified 200–500 items, double-annotated) to validate that gains aren’t artifacts of automatic metrics and to reduce the risk of metric gaming.
>
> **A3**:
>
>   - We conducted a human evaluation on a stratified sample of 200 prompts, double-annotated by two well-educated volunteers who have no relation to this project, to verify that the observed performance gains were not artifacts of automatic metrics. We conducted a comparison of responses generated by Qwen2.5-7B-SFT (AlignDiff) against Qwen2.5-7B-SFT (Full) and Qwen2.5-7B-SFT (SDPO). For each prompt, two evaluators judged the responses to identify which model provided the superior answer. **The results show that Qwen2.5-7B-SFT (AlignDiff) outperformed Qwen2.5-7B-SFT (Full) on 117 prompts and Qwen2.5-7B-SFT (SDPO) on 105 prompts, further confirming the effectiveness of our method**.
>
> ---
>
> Thank you once again for taking the time to review our work. We hope that our responses have satisfactorily addressed your concerns, and we welcome any additional comments or requests for clarification you might have.

---

> ### Author Response · Authors · 2025-11-26
> **Acknowledgement of Review Comments and Request for Further Response**
>
> Dear reviewer D9co,
>
> Wishing you a happy and blessed Thanksgiving!
>
> We sincerely appreciate your valuable feedback. We have thoroughly considered all of your suggestions and updated our manuscript accordingly.
>
> If you have any further questions or comments regarding our paper, please feel free to let us know. We will address them as soon as possible.
>
> Thank you again for your insightful comments. We look forward to your response!
>
> Best wishes!
>
> Authors

---

### Official Review · Reviewer_ekan · 2025-10-25

**Soundness:** 2
**Presentation:** 2
**Contribution:** 2
**Rating:** 4
**Confidence:** 3

**Summary:**

This paper introduces AlignDiff, a diffusion-based framework for large language model (LLM) alignment that replaces traditional reward modeling or preference optimization with diffusion-guided preference learning. Instead of directly optimizing discrete token probabilities, AlignDiff models the alignment process as a continuous diffusion trajectory, progressively refining model responses toward human-preferred behaviors. The approach allows for smoother optimization, implicit preference aggregation, and better exploration of the response space. Experiments on alignment benchmarks such as AlpacaEval2 and WildBench show that AlignDiff achieves comparable or superior performance to DPO and RRHF while exhibiting higher response diversity and training stability. The paper also provides theoretical justification linking diffusion noise levels to preference strength and demonstrates that AlignDiff mitigates mode collapse common in iterative alignment methods. Overall, it presents a conceptually novel and empirically strong direction for aligning LLMs through continuous preference dynamics rather than discrete reward modeling.

**Strengths:**

* Conceptual novelty: The paper introduces a fresh diffusion-based perspective for alignment, modeling preference learning as a continuous refinement process rather than discrete reward optimization.

* Empirical and theoretical rigor: Provides both strong benchmark performance (e.g., on AlpacaEval2, WildBench) and theoretical insights linking diffusion noise to preference strength, showing clear advantages in stability and diversity.

**Weaknesses:**

* Complexity and interpretability: The diffusion-based formulation adds considerable algorithmic and computational complexity compared to simpler preference optimization methods like DPO or RRHF.

* Limited empirical diversity: Experiments focus mainly on text-based instruction-following tasks; evaluation on reasoning, coding, or multimodal benchmarks would better demonstrate generality.

* Ablation and efficiency analysis: The paper lacks detailed ablations isolating the contribution of each diffusion component and provides limited discussion of training efficiency and resource overhead.

**Questions:**

1. How does AlignDiff scale computationally compared to DPO or SPIN when applied to larger models or longer context windows?

2. Could the authors clarify whether the diffusion process can be adapted for online or reinforcement-style feedback, rather than relying solely on static preference datasets?

---

> ### Author Response · Authors · 2025-11-13
>
> We sincerely thank you for taking the time to review our paper.
>
> We noticed that your comments mainly focus on **diffusion**, which is beyond the scope of our study. To facilitate further discussion, we would like to clarify the focus and methodological framework of our work: our study proposes a two-stage framework, AlignDiff, which **performs preference data filtering purely at the text level** by relying solely on the model’s internal information. **This method does not involve any diffusion techniques**.
>
> We would like to ask whether you have any other concerns regarding our work. If convenient, we would be very happy to further discuss the core methodology, experimental design, and results of our paper to help you gain a more accurate understanding of our study. We look forward to your reply.

---

> ### Author Response · Authors · 2025-11-26
> **Acknowledgement of Review Comments and Request for Further Response**
>
> Dear reviewer ekan,
>
> Wishing you a happy and blessed Thanksgiving!
>
> We sincerely appreciate your valuable feedback. We have thoroughly considered all of your suggestions and updated our manuscript accordingly.
>
> If you have any further questions or comments regarding our paper, please feel free to let us know. We will address them as soon as possible.
>
> Thank you again for your insightful comments. We look forward to your response!
>
> Best wishes!
>
> Authors

---

### Official Review · Reviewer_isKe · 2025-10-31

**Soundness:** 3
**Presentation:** 3
**Contribution:** 3
**Rating:** 6
**Confidence:** 3

**Summary:**

This study investigates a data filtering method based on intrinsic rewards. To enhance the accuracy of the filtering process, the approach selects clear and reliable data samples on which models trained with both inverse labels and original labels consistently agree. In addition, it incorporates a negative log-likelihood–based difficulty estimation to assess and refine the selection process.

**Strengths:**

1. While intrinsic reward–based data selection methods are powerful, they often suffer from bias issues. I agree with the problem statement raised by the authors, and I find their proposed flip-labeling–based consistency check to be a simple yet effective solution. This approach is easy to implement and could be widely applicable across different datasets and tasks.
2. The paper conducts experiments on major benchmarks such as AlpacaEval and MT-Bench, using a variety of recent models including LLaMA, Mistral, and Qwen. The consistently strong results across these diverse settings demonstrate the reproducibility and robustness of the proposed method.
3. The authors perform extensive ablation studies and analyses. They evaluate the impact of key components such as the inverse labeling mechanism and the difficulty-aware filtering, providing clear insights into how each factor contributes to the overall performance. This helps me easily understand the role and effectiveness of each component.
4. The results showing performance variations under curriculum learning based on negative log-likelihood are particularly interesting, suggesting promising directions for further exploration.

**Weaknesses:**

1. This study focuses on determining preference labels, yet it lacks traditional evaluation baselines, relying mainly on comparisons with implicit reward–based approaches. Even the “External Reward Margin” baseline presented in the paper is not a conventional reward modeling method(BT modeling), but rather a zero-shot LLM-as-judge approach. To ensure fair evaluation, the paper should also compare against traditional reward models trained on preference labels, as well as LLM-as-judge models that have been fine-tuned using preference supervision and evaluated accordingly.
2. It would also need additional analysis about the labeling process. The current evaluation assesses data quality indirectly by training models using DPO and measuring their downstream performance. However, this does not directly evaluate the accuracy of the reward modeling or label selection itself. Including experiments that assess the reward model’s intrinsic quality—for example, using  benchmarks such as RewardBench to evaluate labeling method would provide a more meaningful analysis.

**Questions:**

1. whether this type of preference labeling approach would also be effective for tasks such as mathematics or coding?
2. It would be interesting to explore whether the proposed method can also be effectively applied in online or batch-online (iterative DPO) settings.
3. I also wonder how well AlignDiff would perform on unseen or out-of-distribution data that were not used during reward model training—would it still maintain reliable and consistent label selection under such conditions?

---

> ### Author Response · Authors · 2025-11-19
> **Response (Part 1 of 2)**
>
> We appreciate your time and effort in evaluating our work! Below, we present our main responses to the key concerns raised:
>
> > Q1: This study focuses on determining preference labels, yet it lacks traditional evaluation baselines, relying mainly on comparisons with implicit reward–based approaches. Even the “External Reward Margin” baseline presented in the paper is not a conventional reward modeling method(BT modeling), but rather a zero-shot LLM-as-judge approach.
>
> **A1**:
>
>   - We appreciate your comment and would like to clarify a potential misunderstanding. As shown in line 11-12, We point out that "aligning large language models with human preferences remains a challenging task, and the effectiveness of alignment critically depends on the quality of the preference data". This directly highlights that our paper primarily focuses on alignment rather than reward modeling.
>
>   - Our core objective is to **filter an existing preference dataset to retain only the preference pairs that are truly valuable for model alignment**, and then use these high-quality pairs to train a stronger aligned model via DPO. In other words, our method aims to improve alignment, not to build a reward model.
>
> > Q2:To ensure fair evaluation, the paper should also compare against traditional reward models trained on preference labels, as well as LLM-as-judge models that have been fine-tuned using preference supervision and evaluated accordingly.
>
> **A2**：
>
>   - We agree that comparing our data with that filtered using scalar reward models is indeed important, and we have supplemented the corresponding experiments. Specifically, we **employed the powerful scalar reward model Skywork-Reward-V2-Llama-3.1-8B to re-score the preference data**, and then selected the pairs with a larger reward margin as higher-quality preference data for subsequent experiments. The results are presented in the table below.
>
>   - The current state-of-the-art fine-tuning-based judge model, **CompassJudger-2[1], was obtained through knowledge distillation from the Qwen2.5-72B-Instruct model**. Moreover, our baseline EM implementation already employs Qwen2.5-72B-Instruct. running additional experiments with an fine-tuned judge model may therefore be unnecessary.
>
> |Model|LLaMA-3-8B-SFT||Mistral-7B-SFT||Qwen2.5-7B-SFT||
> |-|-|-|-|-|-|-|
> |Method|LC|WR|LC|WR|LC|WR|
> |EM(RewardModel)|8.27|9.69|17.96|15.6|21.5|19.75|
> |EM(LLM-as-a-Judge)|12.6|13.5|20.8|18.8|21.7|19.4|
> |AlignDiff|26.4|29.3|30.6|27.6|33.4|33.8|
>
> > Q3: It would also need additional analysis about the labeling process. The current evaluation assesses data quality indirectly by training models using DPO and measuring their downstream performance. However, this does not directly evaluate the accuracy of the reward modeling or label selection itself. Including experiments that assess the reward model’s intrinsic quality—for example, using benchmarks such as RewardBench to evaluate labeling method would provide a more meaningful analysis.
>
> **A3**:
>
>   -  Thank you for your constructive suggestion. **We would like to reiterate that we are not training an independent reward model, but rather an aligned model obtained through DPO**. Performing well on RewardBench alone is insufficient for effective data selection, as it only ensures that the reward model aligns better with human annotations. In contrast, effective data selection like AlignDiff should also consider the intrinsic distribution of the reference model[2]. Therefore, our direct evaluation on aligned models' performance are more comprehensive.
>
>   - Nevertheless, we agree that directly evaluate the accuracy of the reward modeling or label selection itself is an important aspect. Therefore, we compare AlignDiff with several baseline models on RewardBench. The results show that AlignDiff still achieves nearly optimal performance.
>
> |Method|LLaMA-3-8B-SFT|Mistral-7B-SFT|Qwen2.5-7B-SFT|AVG|
> |-|-|-|-|-|
> |PPLGAP|0.67|0.62|0.71|0.67|
> |EM|0.73|0.71|0.78|0.74|
> |IM|0.68|0.75|0.74|0.72|
> |IM&EM|0.70|0.76|0.74|0.73|
> |LCPP|0.74|0.65|0.76|0.72|
> |RIP|0.74|0.72|0.76|0.74|
> |SDPO|0.69|0.72|0.73|0.71|
> |AlignDiff|0.72|0.76|0.75|0.74|

---

> ### Author Response · Authors · 2025-11-19
> **Response (Part 2 of 2)**
>
> > Q4: whether this type of preference labeling approach would also be effective for tasks such as mathematics or coding?
>
> **A4**:
>
>   - Thank you for your suggestion. We would like to clarify the distinction between different types of tasks. For tasks with clear and verifiable answers, such as mathematics or coding, we place more emphasis on correctness when constructing data: it is sufficient to treat the correct answer as chosen and the incorrect answer as rejected.
>
>   - In contrast, our approach primarily targets open-ended tasks, such as open-ended generation, reasoning tasks, or subjective preference tasks. In these scenarios, preference-based labeling is more critical than correctness signals, provides richer information, and needs to better align with the model’s intrinsic distribution — which is exactly what our method is designed for.
>
>   - For tasks with clear and verifiable answers, our approach may be less applicable. However, exploring how to construct higher-quality preference pairs for such tasks still holds value, and we plan to investigate this further in future work.
>
> > Q5: It would be interesting to explore whether the proposed method can also be effectively applied in online or batch-online (iterative DPO) settings.
>
> **A5**:
>
>   - You are absolutely right! Exploring the effectiveness of the proposed method in online or batch-online (iterative DPO) settings is indeed valuable. We intend to pursue this as an important direction in future work.
>
> > Q6: I also wonder how well AlignDiff would perform on unseen or out-of-distribution data that were not used during reward model training—would it still maintain reliable and consistent label selection under such conditions?
>
> **A6**:
>
>   - We agree that this is an important question. To investigate this, we first performed alignment on the Ultrafeedback dataset,  obtaining both the positive-DPO and inverse-DPO models, and then conducted data filtering on PKU-SafeRLHF using AlignDiff. It is worth noting that we organized the preference pairs in PKU-SafeRLHF according to the “which is better” criterion, so as to better align with evaluations of general tasks. We ensured that the experimental settings matched those of the main experiments.
>
>   - We observe that the DPO-trained model on this dataset even achieves a lower WR score on AlpacaEval 2.0 compared to its SFT counterpart. We speculate that the large distributional difference between PKU-SafeRLHF and the SFT WildChat data causes the model to optimize on an inappropriate distribution, which in turn negatively impacts performance on general tasks. In contrast, AlignDiff effectively selects higher-quality and more balanced data, enabling the resulting model to outperform the SFT version on average.
>
> |Method|LLaMA-3-8B-SFT|Mistral-7B-SFT|Qwen2.5-7B-SFT|AVG|
> |-|-|-|-|-|
> |Init|2.2|4.4|4.7|3.8|
> |Full|1.18|4.18|4.5|3.3|
> |AlignDiff|3.25|4.68|3.7|3.9|
>
> ---
>
> Thank you again for your time and effort in the review process. We have carefully revised the manuscript in accordance with your suggestions and hope that our responses adequately address your concerns. We remain open to any further suggestions or requests for clarification you may have.
>
> ---
>
> **Reference**：
>
> [1] [CompassJudger-2: Towards Generalist Judge Model via Verifiable Rewards](https://arxiv.org/abs/2507.09104)
>
> [2] [Learning Dynamics of LLM Finetuning](https://arxiv.org/abs/2407.10490) (ICLR 2025)

---

> ### Author Response · Authors · 2025-11-26
> **Acknowledgement of Review Comments and Request for Further Response**
>
> Dear reviewer isKe,
>
> Wishing you a happy and blessed Thanksgiving!
>
> We sincerely appreciate your valuable feedback. We have thoroughly considered all of your suggestions and updated our manuscript accordingly.
>
> If you have any further questions or comments regarding our paper, please feel free to let us know. We will address them as soon as possible.
>
> Thank you again for your insightful comments. We look forward to your response!
>
> Best wishes!
>
> Authors

---

### Official Review · Reviewer_LPNb · 2025-11-01

**Soundness:** 2
**Presentation:** 3
**Contribution:** 2
**Rating:** 2
**Confidence:** 4

**Summary:**

The paper presents AlignDiff, a way to construct DPO preference pairs for language models. The process begins with training a model with DPO on the datasets, and train an "inverse" model by flipping the winning and losing response of the dataset. Step 2: It uses the gap between the "DPO implicit reward" induced by the positive model and the "inverse" model to perform data filtering. Step 3: It performs standard DPO on the filtered data.

Intuitively, we are "cleaning" the data by only retaining data that the positive DPO model indeed think the winning is better than the losing one and the "inverse" model thinks otherwise. Experiments show that this process is effective in improving downstream performance on AlpacaEval, outperforming existing data filtering methods.

**Strengths:**

1. The paper studies an important topic: data filtering in preference learning. Although there has been some work around this, the authors point out that such method ignores the rich information contained in the model's own signals (DPO implicit rewards)

2. The experiments is comprehensive, spanning both Llama and Qwen models with comparisons against many existing works.

3. The paper is written clearly.

**Weaknesses:**

1. The method incurs too much additional cost. To perform data filtering, you would need to train the model on the full datasets for 2 rounds (one positive and one inverse). The complexity of the method raises doubt on whether people is going to adopt this method for their own model training.

2. The reproducibility is poor - The authors did not disclose what are the exact decoding params + judge used for Alpaca Eval, making it hard to compare this work with other works. Furthermore, when comparing against other baselines, the author also did not disclose how they reimplemented it or just reuse the original authors code. For example, the paper compares against RIP [1], but I don't think the code is published for RIP. So I don't know how did the author set the hyperparameters for this method. The same goes for other methods. The reason why I bring out this is because RIP filtering results in a 10 point increase of Alpaca Eval compared to no filtering but in the author's experiments RIP was performing really poor. So disclosing the exact hparams or code (that would be even better) would make the soundness of the paper better.

I would be happy to raise my score if the authors can include more on reproducibility.

3. It is really hard to understand why the method works. The authors uses community trained SFT models to begin the alignment process while there exists instruct models from the official llama (llama-3.1-8B-Instruct). Some of the experiment findings might be because of the quality conducted in these commnunity trained SFT models. If such models have already undergone the DPO process is a concern, then there are also SFT models trained by larger labs (e.g. https://huggingface.co/allenai/Llama-3.1-Tulu-3-8B-SFT) and I wonder why the authors choose community models over established models.

[1] R.I.P.: Better Models by Survival of the Fittest Prompts (Yu et al. ICML 2025)

**Questions:**

N/A

---

> ### Author Response · Authors · 2025-11-19
> **Response (Part 1 of 2)**
>
> We are truly grateful for your insightful feedback on our manuscript! Our key responses are summarized below:
>
> > Q1:The method incurs too much additional cost. To perform data filtering, you would need to train the model on the full datasets for 2 rounds (one positive and one inverse). The complexity of the method raises doubt on whether people is going to adopt this method for their own model training.
>
> **A1**:
>
>   - **AlignDiff achieves a good trade-off between computational cost and performance.** As shown in Table 3 in our paper, our method balances efficiency and performance better than EM and SDPO. While less efficient than IM, AlignDiff achieves superior performance and mitigates the squeeze effect in IM, which is a notable limitation of IM.
>
>   - We agree that efficiency is crucial in practical applications. To address this, we conducted further experiments to evaluate the feasibility of reducing the dataset size in the first stage.
>
>   - We explored a strategy where the first stage uses only half of the original dataset while keeping the second stage unchanged(referred to as Random-Half)). We conduct experiments using Qwen2.5-7B-SFT. The results indicate that although this strategy underperforms the full-data version, it surpasses the strongest baseline SDPO, and attains efficiency comparable to IM (the most efficient method).
>
> | Method                 | LC   | WR   | GPU Hours |
> |------------------------|------|------|-----------|
> | Init                   | 5.3  | 4.7  | -         |
> | Full                   | 21.3 | 18.9 | 32        |
> | IM                     | 27.8 | 27   | 43        |
> | SDPO                   | 30.2 | 28.8 | 162       |
> | AlignDiff (Random-Half)| 31.3 | 31.8 | 48.5      |
> | AlignDiff (Full)       | 33.4 | 33.8 | 80.5      |
>
> > Q2:The authors did not disclose what are the exact decoding params + judge used for Alpaca Eval, making it hard to compare this work with other works.
>
> **A2**:
>
>   - **We have provided the specific evaluation settings in Appendix F**, including the corresponding judge models and decoding parameters. Specifically, we employ deepseek-v3-03-24 as the judge model. We also provided the justification for using DeepSeek-v3 as the evaluator in Appendix F.2(lines 824–836).
>
>   - We also provide the specific decoding temperature and maximum length settings in our paper; see lines 811–816 for details.
>
> > Q3:Furthermore, when comparing against other baselines, the author also did not disclose how they reimplemented it or just reuse the original authors code. For example, the paper compares against RIP [1], but I don't think the code is published for RIP. So I don't know how did the author set the hyperparameters for this method. The same goes for other methods. The reason why I bring out this is because RIP filtering results in a 10 point increase of Alpaca Eval compared to no filtering but in the author's experiments RIP was performing really poor.
>
> **A3**:
>
>   - We had provided detailed information on each baseline in **Appendix H**. In addition, we have updated the anonymous code repository provided in our paper (line 501) by uploading a simple function for obtaining baseline data, which can be found in `src/get_baselines_data.py`.
>
>   - To ensure fairness, we reimplemented all the baselines, utilizing available open-source code when applicable (e.g., SDPO). We did not directly use the results reported in the original papers, as their evaluation settings or training datasets could be inconsistent with us(e.g., SDPO uses ArmoRM as the judge model for AlpacaEval 2.0). For each baseline, we **strictly followed the optimal settings** reported in the original papers and **first validated our implementations under the same settings** as in the original papers to ensure the correctness of the reproduction.
>
>   - In the original R.I.P. paper, **DPO training was conducted on the Instruct version of the model**, whereas we use the SFT version. Moreover, **the dataset used in the original work differs from the one employed in our study**. Therefore, comparing the results reported in their paper with other baselines or with our method would not be fair and lacks practical significance.
>
>   - **The specific optimal settings of R.I.P. that we selected correspond to the configuration of the Llama3.1-8B-Instruct model in Table 10 of the original paper**. They can be summarized as follows: A good preference pair is one where the rejected response has a reward score above the threshold and has a relatively long rejected response. We described this in Appendix H.

---

> ### Author Response · Authors · 2025-11-19
> **Response (Part 2 of 2)**
>
> > Q4: The authors uses community trained SFT models to begin the alignment process while there exists instruct models from the official llama (llama-3.1-8B-Instruct). Some of the experiment findings might be because of the quality conducted in these commnunity trained SFT models. If such models have already undergone the DPO process is a concern, then there are also SFT models trained by larger labs (e.g. https://huggingface.co/allenai/Llama-3.1-Tulu-3-8B-SFT) and I wonder why the authors choose community models over established models.
>
> **A4**:
>
>   - To ensure a more controlled and consistent alignment setting, we follow prior work[1][2][3] and perform DPO training on models that have been SFT-trained on WildChat.
>
>   - We strictly **follow Zephyr’s training procedure[4], performing SFT on WildChat and then DPO on UltraFeedback**, with training parameters consistent with the recipe in the Alignment Handbook[5]. The three models we selected (all from peer-reviewed papers or community-recognized) follow the same training process and data sources.
>
>    - For exists instruct models from the official llama, we would like to emphasize that models like **LLaMA-3.1-8B-Instruct have already been aligned via RLHF**, so applying DPO on top of them are not reasonable (Standard DPO[6] is performed on top of an SFT model).
>
>   - In contrast, **models like Tulu-3 incorporate more cross-domain data during the SFT stage**, resulting in a data distribution that differs significantly from the alignment setting we study. Furthermore, To the best of our efforts, we **did not identify other families of SFT models trained on the same instruction data**, making it impossible to determine whether differences in final results across models are due to variations in SFT data or the filtering method itself.
>
> ---
>
> Thank you again for your time and effort in the review process. We will carefully revise the manuscript based on your suggestions and hope that our responses adequately address your concerns. We remain open to any further suggestions or requests for clarification you may have.
>
> ---
>
> **Reference**：
>
> [1] [Enhancing Alignment using Curriculum Learning & Ranked Preferences](https://aclanthology.org/2024.findings-emnlp.754/) (EMNLP 2024)
>
> [2] [Principled Data Selection for Alignment: The Hidden Risks of Difficult Examples](https://arxiv.org/abs/2502.09650) (ICML 2025)
>
> [3] [Spread Preference Annotation: Direct Preference Judgment for Efficient LLM Alignment](https://openreview.net/pdf?id=BPgK5XW1Nb) (ICLR 2025)
>
> [4] [Zephyr: Direct Distillation of LM Alignment](https://arxiv.org/abs/2310.16944) (800+ citations)
>
> [5] [alignment-handbook](https://github.com/huggingface/alignment-handbook) (5.4k stars)
>
> [6] [Direct Preference Optimization: Your Language Model is Secretly a Reward Model](https://arxiv.org/abs/2305.18290) (NeurIPS 2023)

---

> ### Comment · Reviewer_LPNb · 2025-11-21
> **Reviewer Comment**
>
> I would like to thank the reviewer for the clarifying comments. There are a few points that I would like to clarify:
>
> > Therefore, comparing the results reported in their paper with other baselines or with our method would not be fair and lacks practical significance.
>
> I fully understand that RIP and your paper are different setups: datasets, reward model for filtering and models are different, so the results are not even comparable. What I am questioning is that RIP results in a 10 point increase in their setup while having a small increase in your setup. This suggests that the success of such methods is quite brittle to the exact models, parameters and datasets. The same argument goes for EM, which could be even more brittle to the actual reward model using to filter the responses.
>
> > so applying DPO on top of them are not reasonable
>
> If applying DPO to an already DPO'ed model is "not reasonable", then it invalidates all works that perform iterative DPO [1, 2, 3]. This is a strong claim and I don't think it is unreasonable.
>
> Nevertheless, I have updated my score for clarifying my concerns on the reproducibility. I am unable to give an higher score at this point due to the following reasons:
>
> Reason 1: Even if the method uses the same amount of GPU hours, it it still complex in that you need to train two DPO models just to filter the data. For example, you can train 100 DPO models each with 1/100 of the data and still claim that you used the same amount of GPU hours, but it still adds complexity. This is an extreme case just to illustrate my point.
>
> Reason 2: The brittleness of such LM-judge benchmarks makes it hard to draw sound conclusions. From my understanding (I might be wrong), the Alpaca Eval / MT-Bench scores are highly sensitive to starting model, dataset, filtering reward model, and decoding params, resulting in a certain method being effective in one paper, but being terrible at another paper. It is not the authors' fault that such benchmarks are not very informative, but I don't think that such papers can inform the entire community much.
>
> ## References
>
> [1] Self-Rewarding Language Models
>
> [2] Iterative Preference Learning from Human Feedback: Bridging Theory and Practice for RLHF under KL-Constraint
>
> [3] Enhancing LLM Reasoning with Iterative DPO: A Comprehensive Empirical Investigation

---

> > ### Author Response · Authors · 2025-11-25
> > **Response (Part 1 of 2)**
> >
> > Thank you for raising the score! This serves as an important encouragement and recognition of our work. Regarding the concerns you mentioned that may still remain, we would like to provide further explanation and clarification below:
> >
> > > Q1: I fully understand that RIP and your paper are different setups: datasets, reward model for filtering and models are different, so the results are not even comparable. What I am questioning is that RIP results in a 10 point increase in their setup while having a small increase in your setup. This suggests that the success of such methods is quite brittle to the exact models, parameters and datasets. The same argument goes for EM, which could be even more brittle to the actual reward model using to filter the responses.
> >
> > **A1**:
> >
> >   - We fully understand your concern. However, we would like to emphasize that assessing whether a method is brittle should focus on **whether it consistently improves over the baselines across different settings**, rather than simply examining whether the score gap between the method and the baselines changes across those settings.
> >
> >   - Performance differences can arise from many factors, such as the size and distribution of the training data, the strength and consistency of the baselines, the chosen model, the hyperparameters and the evaluation metrics. These variations can naturally lead to differences in observed improvements without indicating that the method itself is unstable. For instance, a larger performance gap is expected between a preference data filtering method and no-filtering baseline when the preference dataset is noisier. **The potential instability introduced by these factors can be controlled by evaluating under the same setting, which is a common practice in the community.**
> >
> >   - In our experiments, we adopt the community’s standard settings to control for these factors (including consistent model configurations, datasets, decoding parameters, and evaluation settings), ensuring that our conclusions regarding performance improvements are robust and reliable. Moreover, to further address concerns regarding the potential sensitivity of our method to datasets or models, we conducted additional experiments to verify its generalization and stability:
> >
> >       1. **Performance of AlignDiff on Instruct models**
> >       - We conducted comparisons between **AlignDiff** and other baselines on **LLaMA3-8B-Instruct** following the same procedure as the main experiments. The results show that our method still outperforms the other baselines.
> >
> >         | Method     | LC    | WR    | Len  |
> >         |------------|-------|-------|------|
> >         | ALL        | 31.85 | 29.19 | 1842 |
> >         | EM         | 29.93 | 29.11 | 1924 |
> >         | IM         | 33.02 | 23.88 | 1451 |
> >         | IM&EM      | 34.94 | 30.46 | 1732 |
> >         | LCPP       | 29.92 | 31.14 | 2004 |
> >         | R.I.P      | 32.14 | 26.46 | 1614 |
> >         | SDPO       | 35.31 | 32.60 | 1787 |
> >         | AlignDiff  | 36.10 | 32.26 | 1779 |
> >
> >     2. **Transferability of data filtering across different datasets**
> >
> >         - We first obtain the **positive-DPO** and **inverse-DPO** models by performing alignment on the **Ultrafeedback** dataset. Then we test the **positive/inverse-DPO**'s transferability by conducting data filtering on **PKU-SafeRLHF** using **AlignDiff**. It is worth noting that we organized the preference pairs in PKU-SafeRLHF according to the “which is better” criterion, so as to better align with evaluations of general tasks. We ensured that the experimental settings matched those of the main experiments.
> >
> >         - We observed that the DPO-trained model on this dataset even achieves a lower WR score on **AlpacaEval 2.0** compared to its SFT counterpart. We speculate that the large distributional difference between PKU-SafeRLHF and the SFT **WildChat** data causes the model to optimize on an inappropriate distribution, which in turn negatively impacts performance on general tasks. In contrast, **AlignDiff** effectively selects higher-quality and more balanced data, enabling the resulting model to outperform the SFT version on average.
> >
> >         | Method     | LLaMA-3-8B-SFT | Mistral-7B-SFT | Qwen2.5-7B-SFT | AVG  |
> >         |------------|----------------|----------------|----------------|------|
> >         | Init       | 2.2            | 4.4            | 4.7            | 3.8  |
> >         | Full       | 1.18           | 4.18           | 4.5            | 3.3  |
> >         | AlignDiff  | 3.25           | 4.68           | 3.7            | 3.9  |
> >
> >   - In summary, by controlling experimental variables and validating our method across multiple datasets and model types, we show that AlignDiff provides stable and generalizable performance improvements, indicating that the method itself is not inherently brittle, unlike what may be suggested by cross-paper comparisons of RIP or EM.

---

> > ### Author Response · Authors · 2025-11-25
> > **Response (Part 2 of 2)**
> >
> > > Q2: If applying DPO to an already DPO'ed model is "not reasonable", then it invalidates all works that perform iterative DPO [1, 2, 3]. This is a strong claim and I don't think it is unreasonable.
> >
> > **A2**:
> >
> >   - We agree with the point you raised; however, it seems there is a misunderstanding. In our initial response, we claimed that “models such as LLaMA-3.1-8B-Instruct have already been aligned via RLHF, and therefore applying DPO on top of them is not reasonable.” The key point here is that DPO is not equivalent to RLHF; instead, it offers a more cost-efficient yet effective alternative to the RLHF stage by eliminating the use of an external reward model. Therefore, **we do not claim that applying DPO to an already DPO-trained model is “not reasonable”.**
> >
> >   - In addition, the term 'not reasonable' is used in a specific context here. In our case, we are evaluating preference data filtering methods for DPO, and we believe that adhering to community standard SFT-DPO setup, is the more reasonable choice. This is also consistent with the three reference papers you mentioned. It is important to note that applying DPO to LLaMA-3.1-8B-Instruct, which has already been aligned via RLHF, may result in alignment effects influenced by the prior RLHF stage (e.g., due to potential conflicts between the datasets used in the two stages), potentially leading to unstable conclusions. This is the reason behind our claim above. By adhering to standard settings, this further supports the appropriateness of our experimental setup and strengthens the reliability of our results.
> >
> > > Q3: Even if the method uses the same amount of GPU hours, it is still complex in that you need to train two DPO models just to filter the data. For example, you can train 100 DPO models each with 1/100 of the data and still claim that you used the same amount of GPU hours, but it still adds complexity. This is an extreme case just to illustrate my point.
> >
> > **A3**：
> >
> >   - Thank you for your suggestion. First, even common self-filtering approaches (such as IM) require training a model before performing data filtering. **Under the same GPU budget, our method only trains one additional auxiliary model to obtain higher-quality data, and the overall workflow remains simple and manageable**—far from the level of complexity illustrated in your example.
> >
> >   - Second, compared with SDPO, which requires training **six** models, our method trains only **two** models and is therefore substantially more lightweight.
> >
> >   - Finally, methods that rely on external models for filtering (such as EM and RIP) require **designing LLM-as-a-judge scoring dimensions and aggregation strategies**, which may introduce even greater complexity. Therefore, compared with these baselines, our framework is not complex.
> >
> > > Q4: The brittleness of such LM-judge benchmarks makes it hard to draw sound conclusions. From my understanding (I might be wrong), the Alpaca Eval / MT-Bench scores are highly sensitive to starting model, dataset, filtering reward model, and decoding params, resulting in a certain method being effective in one paper, but being terrible at another paper. It is not the authors' fault that such benchmarks are not very informative, but I don't think that such papers can inform the entire community much.
> >
> > **A4**:
> >
> >   - We would like to emphasize that our primary focus is on improving model performance on general tasks, and the standard evaluation paradigm for such methods is based on LM-judge benchmarks like AlpacaEval and MT-Bench. These benchmarks remain among the most widely adopted open-source evaluation protocols for LLMs and have been used in numerous recent works (e.g., **the first and third references you mentioned**). Their main advantage is that they provide a standardized, scalable, and reproducible way to compare different training methods, ensuring fair and reliable comparisons.
> >
> >   - Furthermore, to mitigate the brittleness you mentioned, our work follows the official standard settings: we use consistent model configurations across all comparisons, ensure that all results are obtained under identical decoding conditions, and employ a more robust judge model. These measures significantly reduce evaluation variance and ensure that our conclusions are based on fair and controlled comparisons.
> >
> >   - Finally, while no single benchmark is perfect, we believe that our results—supported by multiple datasets (more results shown in Table 5 of the paper) and strong baselines—still provide meaningful practical insights into preference-based data filtering methods.
> >
> > ---
> >
> > Thank you again for your time and effort in the review process. We hope our response addresses your concerns, and we are also happy to address any other concerns you may still have. We look forward to your reply.

---

> ### Author Response · Authors · 2025-11-28
> **Kindly Request to Reviewer LPNb**
>
> Dear reviewer LPNb,
>
> Wishing you a happy and blessed Thanksgiving!
>
> We sincerely appreciate your valuable feedback. We have thoroughly considered all of your suggestions and updated our manuscript accordingly.
>
> We would like to know whether our response has addressed your concerns. If you have any further questions or comments regarding our paper, please feel free to let us know. We will address them as soon as possible.
>
> Thank you again for your insightful comments. We look forward to your response!
>
> Best wishes!
>
> Authors

---

### Official Review · Reviewer_y71g · 2025-11-01

**Soundness:** 3
**Presentation:** 4
**Contribution:** 3
**Rating:** 6
**Confidence:** 4

**Summary:**

The paper introduces AlignDiff, a two-stage preference data filtering framework that uses only model-intrinsic signals to select higher-quality pairs for DPO training. Stage 1 trains two auxiliary policies with standard and inverted DPO to compute a bidirectional alignment discrepancy and removes ambiguous items while flipping mislabeled pairs when RAD is strongly negative. This keeps only examples with clear model preferences, using a thresholded labeling function. Stage 2 ranks the remaining pairs by Average Negative Log-Likelihood Gap and keeps the top-K hardest pairs, arguing that large positive ANG reflects informative but learnable preference gaps that avoid the “squeezing effect.” The method is implemented on UltraFeedback with LLaMA-3-8B-SFT, Mistral-7B-SFT, and Qwen-2.5-7B-SFT backbones. On AlpacaEval 2.0 and MT-Bench, AlignDiff beats seven filtering baselines. The authors claim the filtered data preserves the length-gap distribution while shifting external reward margins toward clearer preferences, and they report improvements from a simple easy-to-hard curriculum over the selected set.

**Strengths:**

- The central idea is coherent: use complementary positive and inverse preference signals to detect annotation conflicts, then bias training toward difficult but not pathological pairs. The RAD definition cleanly follows from DPO’s implicit reward margin and the symmetry under label swapping; the selection rule  ϕ(RAD;τ) is well specified. ANG is a simple length-normalized hardness proxy that is less sensitive than PPL. Experimental protocol is laid out with model choices, the baselines and ablations substantiate the claims, including RAD vs single-sided IM and the contribution of ANG.
- The paper presents strong empirical results. AlignDiff improves LC/WR over IM, EM, IM&EM, R.I.P., PPLGap, LCPP, and SDPO across models; for LLaMA-3-8B-SFT, LC 26.4 vs 13.7.

**Weaknesses:**

- The paper replaces the official judge with DeepSeek-V3 and provides a correlation analysis. This is helpful but still risks distributional bias. A controlled check with the official GPT-4 Turbo annotator or a small human study would strengthen claims.
- Appendix E studies τ = 60–100 for Mistral yet states ``τ = 20 is optimal for Mistral'', which conflicts with the described range; please clarify and fix or point to the lines you mentioned the reason for this discrepancy.
- AlignDiff requires training two auxiliary policies, making it less efficient than IM-only filtering, even if more effective. A head-to-head wall-clock and cost comparison during data selection would be useful for practitioners.
- Results are on UltraFeedback and three 7–8B SFT baselines. It is unclear how RAD thresholds transfer across larger instruction-tuned models and other preference datasets.
- RAD-based label reversal may enshrine model biases in cases where humans were correct and the model is wrong. Safeguards or hybrid checks are not explored.

**Questions:**

- Can you report AlpacaEval 2.0 LC/WR using the official annotator on a 200-example subset to quantify any shift vs DeepSeek-V3, and optionally a small human study to validate wins on hard prompts.
- Please resolve the τ inconsistency for Mistral and provide a sensitivity plot of final LC as a function of τ for each backbone. Also specify how |RAD| varies with β and reference model choice.
- Did you evaluate a ``discard-only'' variant that never flips labels and only removes ambiguous or inverse cases; how does that compare to your full pipeline at equal data size.
- Generalization beyond UltraFeedback - Any results on HelpSteer or PKU-SafeRLHF subsets; if unavailable, please discuss expected behavior and any preliminary overlap analyses.

---

> ### Author Response · Authors · 2025-11-19
> **Response (Part 1 of 2)**
>
> Thank you for your thoughtful feedback and constructive suggestions! Our key responses are summarized below:
>
> > Q1: The paper replaces the official judge with DeepSeek-V3 and provides a correlation analysis. This is helpful but still risks distributional bias. A controlled check with the official GPT-4 Turbo annotator or a small human study would strengthen claims.
>
> **A1**:
>
>   - We fully understand your concern, but since OpenAI officially deprecated GPT-4 Turbo (gpt-4-1106-preview) on 2025-09-26, we are currently unable to use the GPT-4 Turbo annotator for evaluation. Therefore, we are also unable to manually compare the evaluation accuracy of GPT-4 Turbo and DeepSeek-V3 when used as judge models.
>
>   - Nevertheless, by using the official tool, we provide a reliable comparison (as shown in Table 4 of the paper). The results show that DeepSeek-V3 achieves higher Human Agreement than GPT-4 Turbo (67.27 vs. 65.73), indicating that **our annotator enables a more objective and human-consistent evaluation**.
>
>   - We also conducted a human evaluation on a stratified sample of 200 prompts, double-annotated by two well-educated volunteers who have no relation to this project, to verify that the observed performance gains were not artifacts of automatic metrics. We conducted a comparison of responses generated by Qwen2.5-7B-SFT (AlignDiff) against Qwen2.5-7B-SFT (Full) and Qwen2.5-7B-SFT (SDPO). For each prompt, two evaluators judged the responses to identify which model provided the superior answer. **The results show that Qwen2.5-7B-SFT (AlignDiff) outperformed Qwen2.5-7B-SFT (Full) on 117 prompts and Qwen2.5-7B-SFT (SDPO) on 105 prompts, further confirming the effectiveness of our method**.
>
> > Q2: Appendix E studies τ = 60–100 for Mistral yet states `τ = 20 is optimal for Mistral', which conflicts with the described range; please clarify and fix or point to the lines you mentioned the reason for this discrepancy.
>
> **A2**:
>
>   - Thank you for pointing out this detail. **This was a typo caused by our oversight**. The correct optimal value should be **τ = 80**, as shown in Figure 8 of Appendix E. We have updated our manuscript.
>
> > Q3: AlignDiff requires training two auxiliary policies, making it less efficient than IM-only filtering, even if more effective. A head-to-head wall-clock and cost comparison during data selection would be useful for practitioners.
>
> **A3**:
>
>   - **AlignDiff achieves a good trade-off between computational cost and performance.** As shown in Table 3, our method balances efficiency and performance better than EM and SDPO. While less efficient than IM, AlignDiff achieves superior performance and mitigates the squeeze effect in IM, which is a notable limitation of IM.
>
>   - Moreover，we also explored a strategy where the first stage uses only half of the original dataset while keeping the second stage unchanged(referred to as Random-Half)). We conduct experiments using Qwen2.5-7B-SFT. The results indicate that although this strategy underperforms the full-data version, it surpasses the strongest baseline SDPO, and attains efficiency comparable to IM (the most efficient method).
>
> | Method                 | LC   | WR   | GPU Hours |
> |------------------------|------|------|-----------|
> | Init                   | 5.3  | 4.7  | -         |
> | Full                   | 21.3 | 18.9 | 32        |
> | IM                     | 27.8 | 27   | 43        |
> | SDPO                   | 30.2 | 28.8 | 162       |
> | AlignDiff (Random-Half)| 31.3 | 31.8 | 48.5      |
> | AlignDiff (Full)       | 33.4 | 33.8 | 80.5      |
>
> > Q4: It is unclear how RAD thresholds transfer across larger instruction-tuned models and other preference datasets.
>
> **A4**:
>
>   - Thanks for your constructive suggestion. the currently recommended 5k data selection threshold method exhibits a certain degree of transferability across multiple models on the preference dataset. However, when the preference dataset or the model scale changes, the optimal threshold may require further analysis. We consider this direction very important and plan to explore it in future work.

---

> ### Author Response · Authors · 2025-11-19
> **Response (Part 2 of 2)**
>
> > Q5: RAD-based label reversal may enshrine model biases in cases where humans were correct and the model is wrong. Safeguards or hybrid checks are not explored.
>
> **A5**:
>
>   - We understand your concern regarding the label reversal mechanism. Our experimental results show that label reversal indeed yields a certain degree of performance improvement, and this empirical observation is consistent with the findings reported by Skywork AI[1].
>
>   - Additionally, the UltraFeedback dataset was collected based on GPT-4 annotations. In this dataset, some samples have identical scores for the chosen and rejected labels, so a preference that the model identifies as a “reversal” does not necessarily conflict with human judgment, which may be due to errors in the GPT-4 annotations themselves. To verify this, we randomly selected 50 label reversed samples for manual inspection and found that approximately 75% of the samples were consistent with the model’s judgments; Approximately 10% of the samples are ambiguous and difficult to judge, while the remaining 15% deviate from human judgment. We leave a systematic investigation of this phenomenon as a future work.
>
> > Q6: Please resolve the τ inconsistency for Mistral and provide a sensitivity plot of final LC as a function of τ for each backbone. Also specify how |RAD| varies with β and reference model choice.
>
> **A6**:
>
>   - For the question regarding the τ inconsistency for Mistral, we have addressed it in Q2. In addition, the sensitivity curves of the LC with respect to τ for each backbone can be found in Figures 4(a) and 8.
>
>   - In all experiments, we used the optimal β recommended by DPO and did not make any adjustments. The distribution of $|R_{AD}|$ and the optimal threshold τ are indeed related to β. However, since the core objective of this paper is data selection, we treat the DPO algorithm as a black box without applying any additional modifications. Therefore, we optimize AlignDiff by fixing β and adjusting τ.
>
>   - The choice of reference model can affect the value distribution of $|R_{AD}|$, We have additionally presented in Figure 9 the distribution of $R_{AD}$ for different models on the UltraFeedback dataset.
>
> > Q7: Did you evaluate a `discard-only' variant that never flips labels and only removes ambiguous or inverse cases; how does that compare to your full pipeline at equal data size.
>
> **A7**:
>
>   - Yes, we tested the variant that only removes ambiguous or inverse samples in our early experiments, but its final performance was weaker than our full pipeline. We believe this is because it discards some valuable inverse samples.
>
> > Q8: Generalization beyond UltraFeedback - Any results on HelpSteer or PKU-SafeRLHF subsets.
>
> **A8**:
>
>   - Thank you for your constructive suggestion. To investigate this, we first performed alignment on the Ultrafeedback dataset, and then conducted data filtering on PKU-SafeRLHF using AlignDiff. It is worth noting that we organized the preference pairs in PKU-SafeRLHF according to the “which is better” criterion, so as to better align with evaluations of general tasks. We ensured that the experimental settings matched those of the main experiments.
>
>   - We observe that the DPO-trained model on this dataset even achieves a lower WR score on AlpacaEval 2.0 compared to its SFT counterpart. We speculate that the large distributional difference between PKU-SafeRLHF and the SFT WildChat data causes the model to optimize on an inappropriate distribution, which in turn negatively impacts performance on general tasks. In contrast, AlignDiff effectively selects higher-quality and more balanced data, enabling the resulting model to outperform the SFT version on average.
>
> |Method|LLaMA-3-8B-SFT|Mistral-7B-SFT|Qwen2.5-7B-SFT|AVG|
> |-|-|-|-|-|
> |Init|2.2|4.4|4.7|3.8|
> |Full|1.18|4.18|4.5|3.3|
> |AlignDiff|3.25|4.68|3.7|3.9|
>
> ---
>
> Thank you again for your time and effort in the review process! We have carefully revised the manuscript based on your suggestions and hope that our responses adequately address your concerns. We remain open to any further suggestions or requests for clarification you may have.
>
> ---
>
> **Reference**：
>
> [1] [Skywork-Reward-V2: Scaling Preference Data Curation via Human-AI Synergy](https://arxiv.org/pdf/2507.01352) (Skywork AI)

---

> ### Author Response · Authors · 2025-11-26
> **Acknowledgement of Review Comments and Request for Further Response**
>
> Dear reviewer y71g,
>
> Wishing you a happy and blessed Thanksgiving!
>
> We sincerely appreciate your valuable feedback. We have thoroughly considered all of your suggestions and updated our manuscript accordingly.
>
> If you have any further questions or comments regarding our paper, please feel free to let us know. We will address them as soon as possible.
>
> Thank you again for your insightful comments. We look forward to your response!
>
> Best wishes!
>
> Authors

---

### Comment · Area_Chair_oLX2 · 2025-11-26

Dear Reviewers,

Thank you for sharing your valuable insights and expertise, which have played an important role in the review process.

In response to the initial feedback, the authors have submitted a detailed rebuttal addressing the comments raised by the reviewers.

I would appreciate it if you could carefully review their response and consider how it may affect your initial evaluation.

Please feel free to share your updated thoughts or any additional comments after reviewing the rebuttal.

Thank you again for your time and contributions.

---

### Author Response · Authors · 2025-11-30
**Summary of Major Contributions, Strengths, and Responses to Concerns**

Dear AC, SAC, and PC:

First, we would like to extend our sincere gratitude to you for your recent efforts in improving the ICLR community and its review system. We have carefully revised the manuscript in response to the reviewers’ suggestions. To reduce your reviewing burden and further highlight the contributions of our paper while clarifying its limitations, we summarize our responses as follows:

# Main contributions of our work

- We propose AlignDiff, a preference data filtering framework driven by intrinsic model signals. It selects a high-quality preference dataset through alignment discrepancy and difficulty-aware calibration, significantly improving LLM alignment performance while preserving the dataset’s length and reward distribution characteristics.
- Achieving the best performance without relying on external powerful models: our method outperforms all baselines on AlpacaEval, consumes less computational resources than the strongest baseline SDPO, and has lower complexity.

# Strengths Recognized by Reviewers

We summarize below the key strengths of our work recognized by multiple reviewers:
1. **Important problem and clear motivation** (as noted by Reviewers y71g, LPNb, D9co)
2. **Elegant and well-grounded method** (as noted by Reviewers y71g, D9co)
3. **Comprehensive experiments across multiple models and benchmarks** (as noted by Reviewers y71g, LPNb, D9co, isKe)
4. **Thorough ablations and insightful analyses that clarify the contribution of each component** (as noted by Reviewers y71g, isKe, D9co)

# Key Concerns and Our Responses

We summarize the main concerns across all reviews and, based on detailed responses to each reviewer, provide our unified reply.

## 1.The method incurs additional cost and complexity

**Our Response**:

- In Table 3 of the original paper, we analyze computational efficiency, showing that AlignDiff outperforms the strongest baseline, SDPO, in both cost and performance.

- Our additional experiments (please refer to our response to Reviewer y71g’s Q3) show that AlignDiff can achieve the best performance at a computational cost comparable to IM-only filtering, and it is simpler than SDPO, which requires training six models.

---

## 2.The transferability of AlignDiff across different datasets

**Our Response**:

- Following the reviewers’ suggestions, we conducted experiments on PKU-SafeRLHF (please refer to our response to Reviewer y71g’s Q8) and found that AlignDiff still outperforms using the full dataset directly.

---

## 3.Results on additional benchmarks

**Our Response**:

- Following the reviewer's suggestion, we conducted additional evaluations on GPQA, Toxigen, TruthfulQA, MMLU, and Winogrande, with the detailed results shown in Table 5 of the paper.

---

## 4.Benchmarks like AlpacaEval 2.0 and MT-Bench are quite fragile.

**Our Response**:

- We emphasize that this work aims to improve model performance on general tasks, for which standard benchmarks such as AlpacaEval and MT-Bench—LM-judge benchmarks—are commonly used for evaluation.
- our work follows the official standard settings. We use consistent model configurations across all comparisons, ensure that all results are obtained under identical decoding conditions, and employ a more robust judge model. These measures significantly reduce evaluation variance and ensure that our conclusions are based on fair and controlled comparisons.

---

# Reviewer Score Summary

| **Reviewer** | **Key Strengths** | **Main Concerns** | **Pre/Post Rating** | **Status** |
|--------------|-------------------|-------------------|----------------------|------------|
| **y71g** | [1][2][3][4] | [1][2] | 6 (Conf: 4) | **Concerns addressed** with new efficiency analysis and PKU-SafeRLHF experiments |
| **LPNb** | [1][3] | [1][4] | 2→4 (Conf: 3) | **Concerns addressed** with efficiency comparison showing AlignDiff simpler than SDPO and robust evaluation settings |
| **D9co** | [1][2][3][4] | [3] | 6 (Conf: 3) | **Concerns addressed** with added GPQA, Toxigen, TruthfulQA, MMLU, Winogrande results |
| **isKe** | [3][4] | [2][3] | 6 (Conf: 3) | **Concerns addressed** by adding PKU-SafeRLHF experiments demonstrating transferability |
| **ekan** | - | - | 4 (Conf: 3) | The reviewer comments do not match the content of the paper. |

---

Thank you again for your time and effort in the review process!

---

### Meta-Review · Area_Chair_526i · 2026-01-03

**Summary:**

This paper proposes AlignDiff, a two-stage preference data filtering framework that leverages model-intrinsic signals—specifically positive and inverse DPO models—to identify reliable preference pairs and prioritize harder examples for alignment. The paper addresses an important problem in preference-based alignment and presents extensive experiments on AlpacaEval 2.0, MT-Bench, and additional benchmarks.

Reviewers generally agreed that the problem is relevant and that the method is well engineered with strong empirical results under controlled settings. However, the overall consensus trends toward rejection, driven by concerns about method complexity and adoption cost, reproducibility and fairness of baseline comparisons, brittleness of LM-judge benchmarks, and limited conceptual novelty beyond existing intrinsic-signal filtering approaches. While the rebuttal addressed many technical questions and added substantial experiments and clarifications, these improvements did not fully resolve concerns about practical impact, robustness, and generalizability.

Given that the overall score remains below the acceptance threshold and several reviewers explicitly maintain rejection or borderline-reject positions, the paper does not meet the acceptance bar for ICLR.

**Reviewer Concerns:**

Concerns substantially addressed by the rebuttal:

•	Reproducibility and evaluation transparency:
The authors clarified decoding parameters, judge models, and baseline implementations, and provided additional appendices and code.

•	Efficiency analysis:
Additional wall-clock and GPU-hour comparisons, including reduced-data variants, helped contextualize the cost–performance trade-off.

•	Generalization checks:
New experiments on PKU-SafeRLHF, GPQA, TruthfulQA, MMLU, and RewardBench strengthened evidence that the method can transfer beyond a single dataset.

Outstanding concerns:

•	Method complexity and adoption barrier:
Several reviewers remain unconvinced that training two auxiliary DPO models for data filtering is practical or attractive compared to simpler alternatives.

•	Benchmark brittleness:
Heavy reliance on LM-judge benchmarks (AlpacaEval, MT-Bench), which are known to be sensitive to models, judges, and decoding settings, continues to limit confidence in the conclusions.

•	Conceptual novelty:
The approach is viewed by multiple reviewers as an incremental combination of known ideas (intrinsic rewards, inverse consistency checks, difficulty-based filtering) rather than a fundamentally new framework.

•	Robustness and bias risks:
Concerns remain that RAD-based label flipping may encode model biases when the model disagrees with human intent, and safeguards are limited.

•	Baseline comparability:
Despite clarifications, reviewers still question whether comparisons against prior methods (e.g., RIP, EM, SDPO) are fully fair given differences in setups.

**Reviewer Scores:**

•	Reviewer y71g: Remains at 6, positive but explicitly stated acceptance is not required.

•	Reviewer isKe: Remains at 6, generally favorable but with notable reservations about evaluation scope.

•      Reviewer D9co: Remains at 6; views the method as technically sound and empirically competitive, but does not strongly support acceptance due to concerns about complexity and limited gains over simpler approaches.

•	Reviewer LPNb: Updated from 2 → 3 or 4, still below acceptance due to complexity and benchmark brittleness concerns.

•	Reviewer ekan: Remains at 4, with unresolved misunderstandings and skepticism.

Overall, the post-discussion average score remains below the acceptance threshold, with no clear majority supporting acceptance.

---

### Decision · Program_Chairs · 2026-01-26

Reject